# The Central Nervous System Modulatory Activities of N-Acetylcysteine: A Synthesis of Two Decades of Evidence

**DOI:** 10.3390/cimb47090710

**Published:** 2025-09-01

**Authors:** Desislava Ivanova Cherneva, Gabriela Kehayova, Simeonka Dimitrova, Stela Dragomanova

**Affiliations:** 1Faculty of Pharmacy, Medical University of Varna, 9002 Varna, Bulgaria; desislavacherneva.2002@gmail.com; 2Department of Pharmacology, Toxicology, and Pharmacotherapy, Faculty of Pharmacy, Medical University of Varna, 9000 Varna, Bulgaria; gabriela.kehayova@mu-varna.bg (G.K.); simeonka.dimitrova@mu-varna.bg (S.D.)

**Keywords:** N-acetylcysteine, neuromodulation, neurotransmission, neuroinflammation, neurodegeneration, neuroprotection

## Abstract

N-acetylcysteine (NAC) has garnered increasing interest for its neurotherapeutic capabilities beyond its recognized functions as a mucolytic agent and an antidote for acetaminophen toxicity. This review consolidates findings from both preclinical and clinical studies to investigate NAC’s diverse modulatory effects on the central nervous system (CNS). NAC primarily functions as an antioxidant by replenishing glutathione and mitigating oxidative stress; however, it produces glutathione-independent effects through the modulation of mitochondrial redox systems, ferroptosis, and the Nrf2-ARE signaling pathway. It plays a significant role in neuroinflammatory processes by inhibiting the production of cytokines, the expression of iNOS, and the activation of microglia. Furthermore, NAC affects various neurotransmitter systems—including glutamatergic, dopaminergic, GABAergic, serotonergic, cholinergic, and adrenergic pathways—by modulating synaptic transmission, receptor activity, and transporter functionality. It promotes neuroprotection through the enhancement of neurotrophic factors, the preservation of mitochondrial integrity, and the upregulation of survival signaling pathways. Recent evidence also emphasizes NAC’s role in gene expression and the regulation of cortisol levels. The extensive range of NAC’s neurobiological effects highlights its therapeutic potential in treating neurodegenerative and neuropsychiatric disorders. Nevertheless, the variability in clinical outcomes indicates a pressing need for more focused, mechanism-based research.

## 1. Introduction

N-acetyl-L-cysteine (NAC), which is an acetylated variant of the amino acid L-cysteine, was first introduced as a mucolytic agent aimed at treating both acute and chronic respiratory ailments [1]. Later, the Food and Drug Administration (FDA) sanctioned its application for acetaminophen overdose, and it is recognized as an essential medicine by the World Health Organisation (WHO) [2]. Beyond these approved indications, NAC’s therapeutic potential has been widely explored, especially in disorders involving oxidative stress [3,4].

Its clinical applications have been comprehensively reviewed by Frye and Berk (2019), Tenório et al. (2021), and Schwalfenberg (2021), covering neurological, psychiatric, and systemic conditions [1,2,5]. Sahasrabudhe et al. (2023) further highlighted NAC’s potential for drug repurposing, outlining its pharmacokinetics and molecular actions, such as redox regulation and anti-inflammatory effects [6]. Nonetheless, the CNS-specific mechanisms of NAC are still only partially explored and warrant further targeted research. Given its favorable safety profile and multimodal actions, NAC has attracted increasing interest as a potential therapeutic agent for CNS disorders [3]. Research over the past decade has emphasized its effects on neuroplasticity [7], neurotransmission, and neuroinflammation [8].

The main component of NAC’s neuromodulatory activity lies in its redox-regulating effects, which are largely mediated through glutathione restoration and buffering of oxidative stress within the CNS. Importantly, NAC reduces neuroinflammation by inhibiting the production of pro-inflammatory cytokines and preventing microglial activation [8,9,10,11]. Moreover, it modulates glutamatergic neurotransmission, lessening excitotoxicity through the regulation of extracellular glutamate via the cystine-glutamate antiporter (xCT) [5,8,12]. NAC also affects dopaminergic signaling, a mechanism particularly relevant in the context of addiction and neurodegenerative disorders [8,13,14,15,16]. It exhibits broader neuromodulatory effects across multiple neurotransmitter systems, including GABAergic, serotonergic, adrenergic, and cholinergic pathways. Additional mechanisms of interest involve its role in glutamine synthesis, modulation of cortisol signaling, and gene regulation relevant to neurotransmission. These multimodal actions of NAC underscore its potential therapeutic applications in psychiatric and neurodegenerative disorders such as schizophrenia, obsessive–compulsive disorder, bipolar disorder, addiction, Parkinson’s disease (PD), and Alzheimer’s disease (AD) [5,8].

Over the past two decades, extensive research has examined NAC’s actions within the CNS. This review synthesizes mechanistic evidence from both preclinical and clinical studies conducted during the last 25 years. Rather than covering general pharmacological effects, we examine how NAC affects neuromodulatory pathways that contribute to its therapeutic effects. Key areas of emphasis include its antioxidant and anti-inflammatory actions, neuroprotection and cell viability, modulation of neurotransmitter systems, and its influence on gene expression. The review is structured thematically, aiming to organize complex findings in a way that highlights specific molecular and cellular pathways relevant to neurological and psychiatric disorders.

## 2. Materials and Methods

This extensive review consolidates literature published since the year 2000, providing a critical assessment of NAC’s neuromodulatory mechanisms and its therapeutic potential in various CNS disorders. By synthesizing results from both preclinical and clinical trials, this review seeks to elucidate NAC’s role in contemporary neurotherapeutics and highlight prospective research avenues for its medical application.

A comprehensive literature search was performed across several scientific databases, including PubMed, Web of Science, ScienceDirect, Scopus, and Google Scholar. The search employed combinations of the following keywords: N-acetylcysteine, NAC, neuromodulation, synaptic plasticity, neuroprotection, neuroinflammation, oxidative stress, glutathione, neurotransmission, and neurodegeneration. Additional relevant studies were discovered through a manual review of the reference lists from pertinent review articles.

The inclusion criteria emphasized the following:(1)Mechanistic studies that explore NAC’s neuromodulatory effects—specifically its impact on oxidative stress, neurotransmission, and neuroinflammatory signaling—in both in vitro and in vivo models;(2)Clinical trials that evaluate the effectiveness of NAC in modulating central nervous system function in neuropsychiatric and neurodegenerative conditions.

## 3. Mechanisms of Central Nervous System Modulation

### 3.1. Antioxidant Properties

NAC is acknowledged for its antioxidant properties and its contribution to redox equilibrium, which is often disrupted in various neurodegenerative diseases [4,17]. In normal physiological conditions, the brain’s antioxidant system efficiently reduces reactive oxygen species (ROS) produced during metabolic processes. Nevertheless, a prolonged oxidative imbalance—characterized by ROS production surpassing the capacity of endogenous antioxidants—leads to molecular alterations, such as protein oxidation and lipid peroxidation. These processes trigger gradual neuronal degeneration and dysfunction [4,17]. NAC alleviates oxidative stress mainly by enhancing intracellular GSH synthesis, with cysteine serving as the rate-limiting precursor [4,18]. The antioxidant effects of NAC are further reinforced by additional mechanisms, which include Nrf2-ARE pathway activation [19,20], direct scavenging of ROS and reactive nitrogen species, detoxifying lipid-derived electrophilic compounds, and boosting the activity of glutathione peroxidase (GPx) and glutathione-S-transferase (GST) [4,21].

NAC’s function in restoring GSH levels is well documented; however, the exact mechanisms that govern its activity in the central nervous system (CNS) are not fully elucidated [9]. These effects encompass both GSH-dependent and independent pathways [6,8].

#### 3.1.1. Glutathione-Dependent Pathways

The antioxidant mechanism of N-acetylcysteine in the brain is primarily attributed to its role as a cysteine donor for glutathione (GSH) synthesis. Glutathione (GSH) is a tripeptide that functions as a crucial redox regulator that maintains oxidative balance and modulates key cellular processes such as neuroinflammation and ferroptosis [22,23,24]. Its various functions encompass the enhancement of antioxidant defense, modulation of neurotransmission, and neuroprotection. A deficiency in GSH levels is linked to neurodegenerative diseases and cognitive decline associated with aging [24]. The intracellular concentration of GSH is maintained through three primary mechanisms:(1)De novo synthesis, where glutamate-cysteine ligase (GCL) and glutathione synthase sequentially synthesize GSH from cysteine, glutamate, and glycine;(2)Redox recycling, in which GPx oxidizes GSH to glutathione disulfide (GSSG), followed by NADPH-dependent reduction back to GSH via glutathione reductase;(3)Xenobiotic conjugation, facilitated by GSTs, which catalyze the covalent attachment of GSH to electrophilic substances [24].

Figure 1 illustrates how NAC contributes to GSH replenishment, highlighting its role as a cysteine precursor.

Preclinical Evidence for GSH replenishment

Preclinical research indicates that NAC-mediated replenishment of GSH offers protection against oxidative damage to neurons [25,26]. This is particularly significant in neurodegenerative disorders such as Parkinson’s disease, where the depletion of GSH intensifies the degeneration of dopaminergic neurons in the substantia nigra (SN) [15,16,27].

The protective role of NAC has been observed across various models of oxidative injury.

For example, Khorchid et al. (2002) illustrated in vitro that exposure to dopamine led to cell death in both oligodendrocyte progenitors and mature oligodendrocytes [28]. This was associated with a notable decrease in intracellular GSH and an elevation in oxidative stress due to the upregulation of heme oxygenase-1 (HO-1). Notably, pre-treatment with NAC inhibited dopamine-induced cell death, restored GSH levels, decreased HO-1 expression, and suppressed the activation of caspase-3 and caspase-9. Additionally, NAC eliminated TUNEL-positive apoptotic cells, underscoring its potent antioxidant and anti-apoptotic properties against oxidative damage [28]. Likewise, in AD cellular model, Wu et al. (2018) demonstrated NAC’s capacity to protect hippocampal neurons from toxicity induced by hydrogen peroxide (H_2_O_2_) through the restoration of GSH and inhibition of the mitogen-activated protein kinase (MAPK) pathway [29]. These findings align with earlier research conducted in 2001 by Olivieri et al., who indicated that NAC pre-treatment countered oxidative damage in SH-SY5Y neuroblastoma cells subjected to H_2_O_2_, UV radiation, or β-amyloid peptides [30]. Despite the inhibition of glutathione reductase by Bis(chloroethyl)nitrosourea, intracellular GSH levels increased. This implies that NAC plays a role in GSH synthesis by facilitating the de novo production of GSH. NAC also seems to affect upstream molecular processes associated with oxidative signaling and protein modification. These findings underscore its possible involvement in the modulation of pathological mechanisms linked to AD, particularly those related to tau pathology and amyloid toxicity.

Furthermore, Dean et al. (2011) examined the capacity of NAC to replenish brain GSH in a rat model that mimicked oxidative stress associated with schizophrenia and bipolar disorder [31]. Their findings indicated that NAC (1000 mg/kg, intraperitoneally) effectively countered GSH depletion caused by 2-cyclohexene-1-one in both saline and amphetamine-treated rats.

In the context of age-related neurodegeneration, NAC also mitigates protein oxidation. Garg et al. (2018) reported notable reductions in ROS, MDA, protein carbonyls, and advanced oxidation protein products in aged rats treated with NAC, which was accompanied by the preservation of neuronal integrity [32].

Going beyond mammalian models, Mocelin et al. (2019) examined a 7-day treatment with NAC aimed at mitigating behavioral changes and oxidative stress caused by an unpredictable chronic stress protocol in a zebrafish model [26]. The researchers evaluated behavioral modifications and analyzed biochemical indicators of oxidative stress, such as ROS, thiobarbituric acid reactive substances (TBARs), nonprotein thiols (NPSHs), along with SOD and CAT activity. NAC treatment successfully countered these stress-related behavioral impairments and oxidative harm, particularly by preventing the reduction in SOD activity, potentially through the enhancement of SOD expression or by scavenging peroxynitrite [26].

In a rodent model of neurotoxicity induced by chemotherapy, NAC also demonstrated neuroprotective properties. Abdel-Wahab & Moussa (2019) found that NAC alleviates damage associated with cisplatin (CIS) by restoring redox balance [33]. Co-treatment with NAC led to a reduction in lipid peroxidation, nitric oxide (NO), and protein carbonyls, while maintaining levels of GSH, total antioxidant capacity, and antioxidant enzymes (GPx, GST, SOD, CAT). Notably, NAC did not produce effects in healthy controls, suggesting that its redox potential is contingent upon the presence of oxidative stress. A recent study by Aremu et al. (2024) further elaborated on these findings in a female rat model, demonstrating that NAC reverses CIS-induced increases in brain kynurenic acid, a metabolite associated with dysfunction in glutamatergic and cholinergic systems, while also doubling the GSH/GSSG ratio and preserving the density of cortical neurons [34]. This dual modulation of redox and neurotransmitter systems positions NAC as a multifactorial adjuvant agent in chemotherapy.

Clinical Findings

Based on encouraging preclinical findings, initiatives have shifted towards clinical trials designed to confirm the neuroprotective capabilities of NAC in human participants. The results of these trials have highlighted both the promise and the translational challenges of NAC therapy. Holmay et al. (2013) examined the impact of NAC on both peripheral and central redox capacity in a cohort of nine non-demented adults, including individuals with PD, Gaucher disease, and healthy controls [35]. Their research provided preliminary clinical evidence indicating that a single intravenous dose of NAC (150 mg/kg) significantly elevated GSH levels in both blood and the occipital cortex across all participants. Nonetheless, the study recognized certain limitations, such as the focus on the occipital cortex rather than PD-relevant areas like the SN, small subgroup sizes that restricted disease-specific interpretations, and the absence of long-term follow-up studies. Despite these constraints, the results offer proof-of-concept for the central antioxidant properties of NAC and highlight the necessity for larger, PD-oriented trials. While Holmay et al. [35] demonstrated that intravenous NAC could enhance central GSH levels, contrasting findings are reported in 2018 by Coles et al. [36]. Their investigation involved administering high-dose oral NAC (6 g/day for 28 days) to individuals with PD (*n* = 5) and healthy controls (*n* = 3). Despite a significant increase in peripheral antioxidant capacity (as indicated by elevated plasma cysteine, GSH/GSSG ratio, and CAT activity), they observed no significant rise in brain GSH levels in the occipital cortex. Additionally, markers of lipid peroxidation, such as MDA and 4-hydroxynonenal (4-HNE), showed no change. Importantly, several participants with PD experienced a temporary exacerbation of motor symptoms at this dosage. These results underscore the challenges posed by NAC’s low oral bioavailability (estimated at 4–10%) due to extensive first-pass metabolism. The limited bioavailability may hinder its capacity to increase central GSH levels, despite an observed enhancement in peripheral antioxidant status.

The 2022 review by Martinez-Banaclocha provides comprehensive insights into the molecular mechanisms underlying neurodegenerative diseases, particularly emphasizing the dysregulation of the cysteine redox proteome [37]. This dysregulation results in oxidative modifications of cysteine residues, which subsequently lead to protein aggregation, a process that is further aggravated by reduced levels of GSH. Conversely, the review also explores the potential of NAC as an antioxidant and a GSH-boosting agent, highlighting its capacity to cross the blood–brain barrier and influence critical cysteine-containing proteins within the brain. It advocates for extensive research involving selected populations over the age of 40 who exhibit diminished blood glutathione levels, comorbidities, and/or mild cognitive impairment, to assess the effects of dietary supplementation with low doses of NAC, which is regarded as a promising and well-tolerated therapeutic option.

#### 3.1.2. Glutathione-Independent Pathways

Direct Antioxidant Effects

NAC’s nucleophilic sulfhydryl (–SH) group can directly neutralize certain electrophiles through redox coupling of its thiol component. However, due to the relatively slow reaction rates and typical concentrations involved, this direct scavenging is improbable for oxidants such as H_2_O_2_, O_2_^•−^, or ONOO^−^, while it remains feasible for more reactive radicals like NO_2_ and hypohalous acids [4,38]. Although these direct antioxidant effects are observable in vitro, their significance in vivo appears limited, stemming from low plasma concentrations of NAC and the prevalence of more abundant endogenous antioxidants such as GSH, ascorbate, and antioxidant enzymes (e.g., SODs, GPs, CAT) [6,38,39]. The primary antioxidant function of NAC in biological systems is believed to be indirect, primarily through the enhancement of intracellular GSH synthesis and the modulation of redox-sensitive signaling. Furthermore, extremely high concentrations of NAC may even present a theoretical pro-oxidant risk by facilitating metal ion–catalyzed radical generation [39]. Therefore, while NAC maintains direct radical-scavenging potential under certain conditions, its overall antioxidant contribution in vivo is likely mediated through GSH-dependent mechanisms rather than through direct scavenging.

Neutralization of lipid peroxides

NAC demonstrates neuroprotective properties by neutralizing harmful lipid peroxides, which is a crucial mechanism in the prevention of ferroptosis during neuronal injury. In research conducted by Karuppagounder et al. (2018), it was found that NAC directly inhibits the formation of toxic lipid peroxides, including 5-hydroperoxyeicosatetraenoic acid, produced by arachidonate 5-lipoxygenase (ALOX5) [21]. This direct neutralization disrupts a fundamental biochemical trigger of ferroptosis. Using both in vitro (hemin-induced ferroptosis in cultured neurons) and in vivo (collagenase-induced intracerebral hemorrhage in mice) models, the researchers illustrated that NAC effectively prevented hemin-induced ferroptotic cell death in cultured neurons and, when administered post-injury in mice, significantly reduced neuronal loss while enhancing functional recovery.

Mitochondrial H_2_S/Sulfane Sulfur Production

A recent study indicates that the antioxidant properties of NAC are linked to a mechanism that involves the metabolism of sulfane sulfur (Figure 1) [40]. Ezeriņa et al. (2018) explored that while NAC acts as a weak direct scavenger of oxidants, it functions as a precursor for the intracellular production of mitochondrial hydrogen sulfide (H_2_S) and sulfane sulfur species, particularly hydropersulfides, which are effective ROS scavengers [40]. By employing redox-sensitive green fluorescent protein 2 in live-cell experiments, the researchers demonstrated that NAC-derived cysteine undergoes enzymatic conversion by 3-mercaptopyruvate sulfurtransferase (MST) and sulfide: quinone oxidoreductase (SQR). The inhibition of these enzymes abolished the antioxidant effect, thereby confirming the enzymatic nature of the pathway. These findings suggest that NAC improves cellular redox balance by producing sulfane sulfur compounds that can directly neutralize ROS. Consequently, NAC facilitates redox homeostasis through a unique and rapid antioxidant mechanism that operates independently of GSH, highlighting its diverse cytoprotective functions [40].

Nrf2-ARE Pathway Activation

Nuclear factor erythroid 2-related factor 2 (Nrf2) serves as a crucial transcription factor that governs antioxidant defenses in response to oxidative stress, for example, resulting from traumatic brain injury (TBI) in rodent models [19]. Under these circumstances, Nrf2 migrates to the nucleus and activates genes driven by the antioxidant response element (ARE), thereby promoting redox homeostasis.

By stimulating the Nrf2-ARE pathway, NAC demonstrates neuroprotective effects, enhancing antioxidant capacity and improving proteostasis through the regulation of the ubiquitin–proteasome system [19,41]. The involvement of NAC in the Nrf2–ARE pathway is further supported by research on its amide derivative, NACA, which mitigated oxidative stress following TBI in rats via Nrf2 activation [20].

NAC potentially activates the Nrf2 pathway through a distinct, GSH-independent mechanism involving hydrogen sulfide (H_2_S) (Figure 2). By acting as a substrate for 3-mercaptopyruvate, NAC increases mitochondrial H_2_S levels, which can persulfonate cysteine residues on Keap1, impairing its ability to sequester Nrf2 in the cytosol. This post-translational modification of Keap1 releases Nrf2, allowing its nuclear translocation and activation of antioxidant gene expression, providing a route for NAC-driven Nrf2 activation. This increases the expression of major antioxidant systems, including GSH, glutathione peroxidase, glutathione S-transferases, thioredoxin (Trx), and thioredoxin reductase (TrxR). The Trx system further sustains Nrf2 activity through a positive feedback loop. Nrf2 also boosts NADPH production by upregulating metabolic enzymes and rerouting glucose metabolism toward the pentose phosphate pathway (PPP). Additionally, it enhances NAD(P)H:quinone oxidoreductase 1 (NQO1) expression, which supports mitochondrial function and lowers oxidative and nitrosative stress [41].

Prior in vitro studies suggest that NAC, through the upregulation of Nrf2, can improve the expression and functionality of the xCT [42,43]. This mechanism enables NAC to diminish glutamatergic tone by enhancing cystine uptake (which serves as a substrate for xCT) and, as a result, facilitating the release of glutamate to stimulate the inhibitory presynaptic metabotropic mGlu2/3 receptor [44]. In a rodent model subjected to chronic alcohol exposure, NAC was found to upregulate Nrf2, resulting in a decrease in hippocampal oxidative stress, as evidenced by a lower GSSG/GSH ratio [44]. This outcome was associated with reduced neuroinflammation and diminished ethanol consumption. The reversal of these benefits upon blocking the xCT antiporter or mGlu2/3 receptors shows that NAC operates through both GSH replenishment and modulation of glutamatergic activity. This dual mechanism not only enhances intracellular antioxidant defenses via cysteine import but also curtails the accumulation of extracellular glutamate [44].

These interrelated actions position Nrf2 at the center of NAC’s redox regulating and neuroprotective capabilities.

Repair of Oxidative DNA Damage

N-acetylcysteine has been shown to provide protective benefits against oxidative DNA damage through the modulation of redox balance and the regulation of critical apoptotic and survival genes [10]. In research conducted by Alam et al. (2019), rats subjected to the organophosphate pesticide fenitrothion for 30 days displayed significant oxidative stress, immune system dysregulation, and neurochemical imbalances [10]. This was demonstrated by increased levels of 8-hydroxy-2′-deoxyguanosine (8OH2dG), a recognized biomarker for DNA oxidative damage, alongside the upregulation of pro-apoptotic genes such as Bax and p53, and a corresponding downregulation of the anti-apoptotic gene B cell lymphoma 2 (Bcl-2). The co-administration of NAC notably reversed these alterations, restoring antioxidant levels (for instance, GSH and SOD), oxidative biomarkers (such as MDA), and reestablishing the equilibrium of apoptotic gene expression. These results suggest that NAC can alleviate genotoxic stress and apoptosis triggered by environmental toxins, thereby underscoring its therapeutic potential in maintaining genomic integrity and neuronal function in oxidative environments [10]. This study confirmed that NAC reduces oxidative damage to cellular macromolecules, which subsequently diminishes the molecular pathways associated with neurodegenerative progression. Importantly, antioxidative mechanisms also interact with inflammatory signaling, highlighting NAC’s role in the modulation of neuroinflammation.

Table 1 presents the mechanisms of antioxidant action of NAC.

### 3.2. Anti-Neuroinflammatory Effects

Inflammation is a fundamental biological response to cellular or tissue damage, infection, or trauma. Although an effective inflammatory response plays a protective role, persistent or excessive inflammatory activity in the central nervous system (CNS) contributes to the pathogenesis of both acute and chronic neurological conditions [48]. Neuroinflammation is widely implicated in the progression of numerous neurodegenerative diseases, such as AD, PD, and multiple sclerosis [49,50].

#### 3.2.1. Suppression of Cytokine Production and Inhibition of NF-κB and iNOS Activities

NAC has been shown to attenuate neuroinflammation by downregulating key pro-inflammatory mediators and signaling pathways, particularly cytokines, NF-κB, and inducible nitric oxide synthase (iNOS) [9,10,11,51,52]. In a rodent model of ischemic stroke, Khan et al. (2004) reported that post-ischemic NAC administration led to significant reductions in cerebral infarct size and tissue damage, while increasing brain GSH levels [51]. Notably, NAC treatment lowered levels of TNF, IL-1β, and iNOS, reduced microglial activation, and prevented apoptosis in the affected brain regions. As demonstrated in the experiment conducted by Khan et al. (2004), a recent study by Kim et al. (2024) confirmed that NAC significantly reduced iNOS expression in the ventral hippocampus of rats exposed to an early-life stress model [53]. This decrease in iNOS is particularly noteworthy, as the enzyme produces considerable amounts of NO during inflammatory conditions, which can combine with superoxide to create highly reactive nitrogen species. By lowering iNOS expression, NAC reduces the production of these reactive species, thus further augmenting its neuroprotective effects.

Another key aspect of NAC’s anti-neuroinflammatory mechanism is its ability to inhibit NF-κB, another transcription factor that governs the expression of various genes associated with inflammation, including those that encode pro-inflammatory cytokines, chemokines, and adhesion molecules [52,54]. By inhibiting NF-κB activation, NAC effectively diminishes the transcription of these inflammatory mediators, thus mitigating the neuroinflammatory cascade [52]. In Abdel-Wahab and Moussa (2019) study, co-treatment with NAC markedly reduced CIS-induced increase in TNF-α, IL-1β, and IL-6 levels in the rats’ brains, indicating that the anti-inflammatory properties of NAC may enhance its antioxidant effects [33].

#### 3.2.2. PPAR-α Activation

A separate anti-inflammatory mechanism involves NAC’s modulation of the peroxisome proliferator-activated receptor alpha (PPAR-α) pathway. This pathway plays a central role in maintaining peroxisomal function and supporting oligodendrocyte (OL) development during inflammatory stress.

In an in vitro model using lipopolysaccharide (LPS)-stimulated glial cultures, Paintlia et al. (2008) observed that pro-inflammatory cytokines secreted by activated glia hinder OL maturation by suppressing peroxisomal protein expression and downregulating PPAR-α activity [55]. The core mechanism consists of an inflammatory cascade: the exposure to LPS initiates the release of pro-inflammatory cytokines, including TNF-α and IL-1β, as well as the production of ROS, which results in a depletion of GSH. As a result, this oxidative stress impairs peroxisomal biogenesis and functionality by downregulating the activity of PPAR-α, directly damaging OL. The findings indicate that NAC can effectively influence PPAR-α activity and improve peroxisomal function.

#### 3.2.3. JAK/STAT Pathway Modulation

The JAK/STAT signaling pathway is crucial in mediating immune and inflammatory responses and is extensively found in various regions of the brain, such as the cerebral cortex and hippocampus. Recent studies suggest an additional mechanism through which NAC exerts anti-neuroinflammatory effects by modulating the JAK/STAT signaling pathway. Al-Samhari et al. (2016) showed that intraperitoneal treatment with NAC led to a down-regulation of STAT3 mRNA expression and protein phosphorylation, while simultaneously enhancing SOCS3 gene expression in a depression rat model [56]. The processes through which NAC reduces STAT3 expression likely involve the inhibition of phosphorylation and tyrosine kinase activity of upstream proteins such as JAK1, JAK2, JAK3, and Tyk2, consequently preventing STAT3 phosphorylation at Tyr705. Moreover, NAC’s well-documented role as a precursor to GSH may contribute to its antioxidant effects, as the glutathionylation of STAT3 can impede its activation via JAK proteins. Furthermore, NAC may also influence cytokine receptors that operate through the JAK/STAT signaling pathway [56].

#### 3.2.4. Microglial Modulation

Microglia, the immune cells prevalent in the CNS, serve a dual function when confronted with pathological challenges, including abnormal stimulation, neurotoxins, infections, or tissue injury. In 2017 Shabab et al. published a comprehensive review of the main neuroinflammatory mediators produced by activated microglia and their signaling pathways as potential therapeutic targets in neurodegenerative processes [48]. Microglial activation can result in pro-inflammatory conditions, characterized by the secretion of pathological factors (such as IL-1β, TNF-α, IL-6, NO, and proteases), or in neuroprotective states, influenced by signals like IL-4, IL-10, IL-13, and transforming growth factor-β (TGF-β) [50].

NAC influences microglial activity in part by modulating connexins and pannexins. As detailed in the extensive review by Caruso et al. (2023), connexins and pannexins create protein channels in the microglial cell membrane, resulting in the development of gap junctions, hemichannels, and pannexons [11]. These channels promote intercellular communication by permitting the release of gliotransmitters such as adenosine triphosphate (ATP) and glutamate, as well as calcium ions (Ca^2+^), which are essential in neuroinflammatory mechanisms [11]. NAC has been demonstrated to modulate these channel proteins, thus affecting microglial communication and neuroinflammatory responses. The authors suggest that NAC’s capacity to influence connexin and pannexin activity represents a novel pathway through which it exerts anti-inflammatory effects in the CNS, separate from its well-known antioxidant functions [11].

Additionally, a preclinical study conducted by Sakai et al. (2023) examined the effects of NAC on microglial inflammation and mortality using both in vitro and in vivo mouse model approaches [57]. The research underscores the dose-dependent complexity of NAC’s effects. NAC inhibited LPS-induced synthesis of pro-inflammatory cytokines (TNF-α and IL-1) and nitric oxide in both mouse (MG6 cell line) and human microglial cells at lower concentrations. Conversely, high concentrations of NAC (≥30 mM) led to the aggregation of cellular TNF-α and resulted in mortality across the microglial cell lines. Similarly, high-dose intraperitoneal NAC injections caused microglial mortality in the brains of mice. The impact was reduced in microglial TNF-α-deficient mice and human primary M2 microglia, indicating that the homeostatic function of TNF-α is associated with this negative effect, regardless of nitric oxide production. These findings reveal that while NAC has anti-inflammatory potential, elevated doses may trigger adverse outcomes in microglia, especially through TNF-α accumulation [57].

NAC’s diverse effects on neuroinflammation and neuronal integrity were examined in a translational study on schizophrenia conducted by Dwir et al. in 2021 [58]. This disorder involves dysfunction of parvalbumin-positive interneurons (PVIs), often linked to redox imbalance, GSH depletion, and neuroinflammation in early life. Authors utilized a Gclm knockout mouse model to investigate if NAC and/or environmental enrichment (EE) could reinstate PVI functionality through the inhibition of the matrix metalloproteinases (MMP9)/RAGE pathway. Sequential treatment with NAC followed by EE effectively restored PVI network integrity, suppressed MMP9/RAGE signaling, and normalized associated cognitive deficits. This preclinical study was complemented by a double-blind, randomized, placebo-controlled clinical trial assessing NAC in patients with early psychosis (EP). Likewise, a 6-month treatment with NAC in EP patients reduced plasma soluble RAGE, increased prefrontal GABA levels, and enhanced cognitive function [58]. Collectively, these findings indicate that a combined approach utilizing pharmacological (NAC) and psychosocial (EE) interventions, even when implemented following an initial insult, has the potential to reverse persistent impairments in PVI/perineuronal net integrity, thereby presenting a promising new therapeutic strategy for individuals at risk of early-life neurodevelopmental adversities.

Earlier work by Woo et al. (2008) substantiates the anti-inflammatory function of NAC on activated microglia [59]. The researchers discovered that by inhibiting ROS, NAC markedly diminished the levels of MMP-3 and MMP-9 in microglial cells stimulated by LPS. These metalloproteinases serve as crucial mediators in the inflammatory responses induced by LPS, regulating the expression of iNOS, pro-inflammatory cytokines, and the activation of nuclear factor kappa B (NF-κB), activator protein 1, and MAPK. By inhibiting the expression of MMP-3 and MMP-9, NAC indirectly suppresses these inflammatory pathways [59]. Additionally, treatment with NAC dose-dependently reduced NO and TNF-α production in LPS-activated microglia, with stronger effects at 5–20 mM. This response was linked to NAC’s antioxidant action, which neutralized ROS and blocked redox-sensitive inflammatory pathways, thereby limiting pro-inflammatory mediator release.

A summary of the described anti-neuroinflammatory mechanisms of action of NAC is presented in Table 2.

### 3.3. Enhancement of Neuroprotection and Cellular Viability

NAC enhances neuronal survival via various interconnected mechanisms that diminish oxidative stress, uphold mitochondrial integrity, modulate cell survival pathways, and preserve synaptic plasticity. For example, Sachdeva et al. (2015) illustrated the effectiveness of NAC in mitigating neurotoxicity induced by sodium tungstate in rats [60]. Their research showed that a 3-month exposure to sodium tungstate (100 ppm in drinking water) led to heightened oxidative stress markers (ROS, TBARs, and GSSG), increased GPx activity, and a decline in GSH and glutathione S-transferase (GST) content. This was accompanied by a reduction in brain concentrations of dopamine, serotonin, norepinephrine, and acetylcholinesterase activity. Importantly, the oral co-administration of NAC successfully reversed all these changes, restoring GSH, reducing oxidative markers, and normalizing neurotransmitter levels. This highlights NAC’s dual function in alleviating oxidative stress and reinstating neurotransmitter equilibrium [60]. Further evidence for NAC’s multifaced neuroprotection comes from a study on transgenic PD model mice by Clark et al. (2010) [25]. They found that while NAC administration temporarily increased GSH levels (weeks 5–7), this effect was not sustained after one year. The research findings indicated that NAC conferred neuroprotection via two separate compensatory mechanisms: (1) the decrease in α-synuclein overexpression, and (2) the modification of NFκB translocation from the nucleus to the cytoplasm, thereby preventing the activation of subsequent pro-inflammatory genes. The persistent activity of striatal tyrosine hydroxylase indicates the preservation of dopaminergic neurons. These findings demonstrate that the therapeutic advantages of NAC go beyond temporary modulation of GSH, targeting PD pathology, including protein aggregation and neuroinflammation.

#### 3.3.1. Initiation of Pro-Survival Signaling

ERK Pathway Activation

NAC activates the Ras-extracellular-signal-regulated kinase (ERK) pathway, which is crucial for neuroprotection against amyloid-beta (Aβ) toxicity. This process involves a complex signaling pathway, which includes the activation of cyclin-dependent kinase 5 (Cdk5) and the expression of Bcl-2, thereby enhancing cell survival and diminishing apoptosis. A study by Hsiao et al. (2008) demonstrated that NAC safeguards cultured cortical neurons from Aβ-induced toxicity by stimulating p35/Cdk5 activity [61]. This neuroprotective mechanism is mediated through a novel pathway involving the sequential activation of several key proteins. NAC application elevated the protein levels of p35, the neuronal-specific activator of Cdk5, thereby enhancing Cdk5 activity. The neuroprotective effects of NAC were lost when Cdk5 activity was pharmacologically or genetically inhibited. Moreover, NAC-induced Cdk5 activation resulted in the phosphorylation and activation of extracellular signal-regulated kinases (ERKs). Blocking the ERK pathway with specific inhibitors also nullified NAC’s protective effects against Aβ toxicity.

This Cdk5-ERK signaling cascade ultimately elevates the expression of Bcl-2, a crucial anti-apoptotic protein. Experiments using Bcl-2 siRNA confirmed that this increase was vital for averting Aβ-induced apoptosis [61]. This pathway also contributes to NAC’s ability to stimulate ERK and Heme oxygenase-1 (HO-1) to mitigate ammonia-induced inflammation [9,61].

Inhibition of MAPK Signaling and Tau Hyperphosphorylation

NAC has demonstrated the ability to inhibit MAPK signaling pathways and diminish tau hyperphosphorylation, revealing novel protective mechanisms that guard against neuronal injury [29]. Initial findings on NAC’s role in MAPK signaling came from Tian et al. (2003), who demonstrated NAC’s inhibition of JNK3 activation in the rat hippocampus during post-ischemic reperfusion, a mechanism linked to scavenging of ROS [46]. Recent studies have built upon these discoveries; Wu et al. (2018) demonstrated that NAC treatment significantly reduced H_2_O_2_-induced tau phosphorylation at multiple sites, including Ser199, Ser202, Thr205, and Ser396 [29]. This effect was facilitated by NAC’s inhibitory action on the MAPK signaling pathway, particularly affecting c-Jun N-terminal kinase (JNK) and p38 MAPK. By preventing tau hyperphosphorylation, NAC aids in maintaining the structural integrity of neurons and inhibits the development of detrimental tau aggregates, thereby protecting neuronal function and survival. Furthermore, Fan et al. (2020) illustrated that NAC safeguards hippocampal neurons against oxidative stress-induced damage in a rat model of depression by inhibiting p38/JNK and concurrently promoting ERK signaling pathways, thereby emphasizing NAC’s significant neuroprotective role through MAPK inhibition [47].

#### 3.3.2. Mitochondrial Protection and Redox Balance

NAC supports neuronal viability by preserving mitochondrial integrity and redox homeostasis. In a rat model of *status epilepticus*, Wang et al. (2024) demonstrated that NAC inhibited the downregulation of parvalbumin (a calcium-binding protein essential for the proper functioning of inhibitory interneurons) by mitigating mitochondrial fission and counteracting oxidative stress [62]. The protective effects of NAC were mediated through two primary mechanisms: the inhibition of CDK5-dynamin-related protein 1 (DRP1) signaling and the modulation of the glutathione peroxidase 1 (GPx1)-nuclear factor kappa B (NF-κB) pathway. DRP1 functions as a crucial regulator of mitochondrial fission, and its hyperactivation under pathological circumstances leads to excessive fragmentation of mitochondria, which in turn results in compromised mitochondrial function and increased levels of ROS. NAC treatment reduced CDK5 activity, thereby decreasing DRP1 phosphorylation and preventing excessive mitochondrial fission. Furthermore, NAC upregulated the expression and activity of GPx1, a principal antioxidant enzyme responsible for detoxifying H_2_O_2_ and lipid peroxides [62]. This contributed to diminished oxidative stress and the inhibition of NF-κB signaling, further aiding in the preservation of parvalbumin expression. NAC also improved mitochondrial morphology, sustained mitochondrial membrane potential, and boosted ATP production, all of which are critical for neuronal survival and function.

A study conducted in 2018 examined the impact of NAC on the functionality of the blood–brain barrier (BBB) using an in vitro mouse model that included brain microvascular endothelial cells and astrocytes [63]. NAC induced time-dependent, cell-specific effects on BBB permeability. In endothelial cells, a 24 h exposure to NAC resulted in the disruption of tight junctions due to a reduction in occludin, delocalization of claudin-5, and an increase in neuronal nitric oxide synthase (nNOS) expression, which led to increased permeability. After 48 h, NAC further compromised tight junction integrity, inhibited endothelial mitochondrial function, and reduced nNOS levels, thus restoring barrier integrity. In contrast, astrocytes displayed different responses; after 24 h, NAC enhanced connexin-37 and 43 gap junctions as well as mitochondrial respiration, while after 48 h, NAC normalized the levels of gap junctions while maintaining barrier stabilization. Notably, the reduction in ROS in astrocytes occurred independently of NOS modulation [63]. The research underscored a crucial mechanistic understanding: the synchronized regulation of mitochondrial activity and junctional proteins surfaced as a fundamental mechanism that explains NAC’s time-dependent effects on BBB permeability. These effects were associated with NAC’s capacity to boost GSH levels, activate mitochondrial complex I, and manage redox balance via NADPH cycling. Collectively, these results indicate that the coordinated regulation of mitochondrial and junctional proteins serves as a vital mechanism for NAC’s temporal influence on BBB functionality.

#### 3.3.3. Modulation of Neurotrophic Factors

NAC modulates neurotrophic factors essential for neurogenesis and the maintenance of synaptic integrity, particularly brain-derived neurotrophic factor (BDNF) [18,64]. BDNF exerts its influence by binding to the tropomyosin receptor kinase B (TrkB), which triggers multiple intracellular signaling pathways, including the phosphatidylinositol 3-kinase (PI3K)/Akt pathway, the Ras/extracellular signal-regulated kinase (ERK) pathway, and the phospholipase C-γ (PLC-γ) pathway. In their comprehensive review, Bavarsad Shahripour et al. (2014) highlighted NAC’s ability to influence the expression of neurotrophic factors, identifying this as one of its neuroprotective mechanisms [18]. The authors observed that treatment with NAC resulted in elevated levels of BDNF that promotes neuronal survival, differentiation, and synaptic plasticity. Lee et al. (2018) provided in vitro validation using SH-SY5Y neuroblastoma cells exposed to cadmium nitrate [64]. Cadmium exposure resulted in decreased phosphorylation of ERK1/2 and compromised mitochondrial function, ultimately leading to apoptosis. Pre-treatment with NAC (1 mM) restored ERK1/2 activity and reduced cell death, suggesting a role in maintaining ERK-dependent neurotrophic signaling in conjunction with redox modulation. This discovery indicates that NAC provides protective benefits not only through redox regulation but also by enhancing neurotrophin signaling, particularly under oxidative or toxic stress.

A summary of the information on the enhancement of Neuroprotection and Cellular Viability by NAC is presented in Table 3.

### 3.4. Modulation of Neurotransmission

#### 3.4.1. Glutamatergic System

NAC exerts its therapeutic effects by modulating the intricately coordinated glutamate signaling system, the disruption of which is implicated in various neuropsychiatric disorders [65]. Proper glutamatergic neurotransmission depends on a delicate balance between the synthesis, release, uptake, and recycling of glutamate. Any dysregulation can lead to conditions such as schizophrenia [66], addiction [67], and autism spectrum disorder [68].

Cystine-Glutamate Antiporter (System xCT) Modulation

One of the primary mechanisms underlying NAC’s neuromodulatory action involves its regulation of the cystine–glutamate antiporter (system xCT), which is predominantly found in astrocytes [8]. This system regulates extracellular glutamate concentrations through non-vesicular release, establishing a vital feedback loop: the released glutamate activates presynaptic group II metabotropic receptors (mGluR2/3), which in turn inhibit vesicular glutamate release and offer protection against N-methyl-D-aspartate receptor (NMDAR)-mediated excitotoxicity (Figure 2) [8,69,70,71]. NAC enhances this pathway by functioning as a cysteine prodrug. Cysteine undergoes dimerization to create cystine, which is subsequently moved into neurons by the cystine-glutamate antiporter [72]. This process elevates extracellular glutamate levels, indirectly influencing both glutamatergic and dopaminergic signaling [73].

Evidence from preclinical studies supports this mechanism. NAC has been shown to provide neuroprotection in models of cerebral ischemia and neurodegeneration [22,74], and to reduce relapse in addiction models. For instance, Kau et al. (2008) demonstrated that NAC administration reduced cocaine-primed reinstatement of drug-seeking behavior in rats [70]. This effect was abolished when system xCT was pharmacologically inhibited in the nucleus accumbens (NAc) using (S)-4-carboxyphenylglycine, directly linking cystine–glutamate exchange to diminished relapse-like behavior. Further supporting this role, Kupchik et al. (2012) investigated the cellular mechanisms by which NAC reduces the propensity for cocaine relapse in rodent models [12]. Building on evidence that relapse is associated with diminished glutamate levels in the NAc core and altered corticostriatal synaptic transmission, the researchers administered NAC via microinjection into the NAc core of rats trained in cocaine self-administration. Additionally, in vitro whole-cell recordings from acute brain slices revealed that NAC displayed a dual, concentration-dependent influence on evoked glutamatergic synaptic currents in the NAc core: lower concentrations diminished amplitude through mGluR2/3 activation, whereas higher concentrations augmented amplitude via mGluR5 activation [12]. Both of these effects were dependent on the uptake of NAC through cysteine transporters and the functioning of the cysteine/glutamate exchanger. Importantly, the inhibition of mGluR5 enhanced the anti-relapse efficacy of NAC, highlighting a potential synergistic therapeutic strategy.

Additional preclinical investigations suggest that NAC may diminish drug-seeking behavior in rodent models by influencing xCT antiporter and glutamate transporter 1 (GLT-1) in NAc and prefrontal cortex [22,75,76].

In the context of schizophrenia, Huang et al. (2018) demonstrated that NAC mitigates glutamatergic dysfunction associated with deficiency in the regulator of G-protein signaling 4 (RGS4) [77]. The deficiency of RGS4 resulted in a diminished expression of SLC7A11, which encodes xCT, thereby impairing the xCT system. This impairment led to a hypofunction of glutamatergic activity in the prefrontal cortex, observed in both brain slices and RGS4 knockdown mice. Notably, treatment with NAC reinstated system xCT activity, normalized glutamate concentrations, and ameliorated behavioral deficits, such as impaired prepulse inhibition and social interactions [77]. NAC alleviates xCT system dysfunction caused by RGS4 deficiency by preserving cysteine and glutamate exchange, offering a targeted method to restore glutamatergic signaling in the prefrontal cortex for this schizophrenia subtype. The study suggests NAC as a promising therapeutic option for RGS4 deficiency-related schizophrenia due to its specific action on the xCT transporter.

Glutamate Transporter 1 (GLT-1) upregulation

Further studies have demonstrated that NAC enhances the expression of glutamate transporter 1 (GLT-1) (Figure 2). By promoting this GLT-1 upregulation and facilitating the clearance of excess synaptic glutamate, NAC contributes to relapse prevention in substance use disorder [76].

Krzyżanowska et al. (2017) examined the modulation of glutamatergic neurotransmission, which is a crucial element in the pathophysiology of cerebral ischemia, by focusing on glutamate transporters [74]. In a rat model of focal cerebral ischemia induced by a 90 min occlusion of the middle cerebral artery (MCAO), the researchers compared the effects of a five-day pretreatment with ceftriaxone (CEF) or NAC to those of ischemic and chemical preconditioning protocols. Both CEF and NAC significantly lowered extracellular glutamate levels in the frontal cortex and hippocampus during episodes of focal cerebral ischemia, achieving effects similar to those of established preconditioning strategies. Immunofluorescence staining for GLT-1 and the xCT subunit of system xCT in astrocytes, neurons, and microglia provided partial mechanistic insight [74]. Specifically, CEF seemed to inhibit the MCAO-induced downregulation of astrocytic GLT-1, while NAC reduced the expression of astrocytic and neuronal xCT.

Additional rodent studies have provided valuable insights into NAC’s neurochemical mechanisms. Moro et al. (2020) examined the efficacy of NAC in promoting the extinction of nicotine-related cues [78]. Prolonged NAC treatment, especially when combined with cue exposure therapy or lever-press extinction, produced lasting anti-relapse effects for up to 50 days following treatment. This notable behavioral enhancement was supported by NAC’s role in restoring essential proteins for glutamate homeostasis in the NAc after a week of treatment, and significantly, by elevating the expression of type II metabotropic glutamate receptors (mGluR2/3) after 50 days. These findings suggest that the capacity of NAC to maintain extracellular glutamate levels is facilitated by enhancing the negative feedback systems that control glutamate release via mGluR2/3. Similarly, in a rat model of diet-induced obesity, Lau et al. (2021) showed that NAC mitigates deficits in astrocytic glutamate transport through enhancement of GLT-1 activity in astrocytes, a process necessary for the removal of surplus glutamate from the synaptic cleft [79]. This mechanism averts the heterosynaptic depression of GABA transmission onto pyramidal neurons, thus preserving GABAergic tone.

This normalization of glutamate homeostasis has also been evidenced in clinical studies involving cocaine-dependent patients diagnosed with schizophrenia, where NAC treatment successfully restored glutamate levels to normal [80].

However, clinical trials assessing its effectiveness in substance use disorders reveal a disparity between mechanistic potential and therapeutic results. A recent meta-analysis conducted by Winterlind, encompassing nine randomized controlled trials, found no notable advantage of NAC over placebo in alleviating cravings across various substance use disorders, despite its well-established modulation of glutamatergic and dopaminergic systems [81]. The meta-analysis identified substantial heterogeneity among trials (I^2^ = 99.26%), influenced by factors such as dosing variability (600–3000 mg/day), differing craving assessment methods (e.g., self-report vs. neuroimaging), and heterogeneous participant characteristics. Subgroup analyses focusing on alcohol and tobacco use disorders similarly failed to demonstrate significant NAC-related benefits, challenging the generalizability of NAC as a universal anti-craving agent. These findings contrast with earlier meta-analyses suggesting more favorable outcomes [82,83], highlighting the need for more standardized and mechanistically aligned clinical trials [81].

NMDA Receptor Modulation

Beyond the xCT system, NAC also plays a role in modulating the activity of NMDAR. While earlier studies suggested a direct binding mechanism, more contemporary research emphasizes NAC’s influence on the receptor’s redox state via GSH [24]. This process entails oxidized GSH’s inhibition of NMDA-mediated calcium influx and its ability to displace ligand binding; however, these interactions seem to depend on the presence of ROS and the receptor’s redox state, rather than occurring under typical physiological conditions [5,8]. These diverse glutamatergic mechanisms collectively underpin NAC’s therapeutic potential, which includes both providing neuroprotection and averting relapse in addiction [22,74].

#### 3.4.2. Dopaminergic System

NAC profoundly influences the dopaminergic system through a wide range of mechanisms, including its antioxidant actions, modulation of mitochondrial function, regulation of dopamine transporter (DAT) activity, influence on dopamine metabolism and turnover, and indirect modulation via the glutamatergic system.

Dopaminergic Protection: Redox Balance, Mitochondrial Function

NAC’s principal neuroprotective mechanism is centered on reducing oxidative stress, and mitochondrial dysfunction that are significant factors in the degeneration of dopaminergic neurons. By effectively scavenging ROS, such as dopamine-derived quinones and H_2_O_2_, NAC safeguards critical elements of dopaminergic neurons from oxidative harm, including mitochondrial complex I, complex IV, and Na^+^/K^+^-ATPase [84].

Cellular Evidence: Redox Restoration and Protection from Dopamine Toxicity

Dopamine toxicity, driven by oxidative stress and the auto-oxidation of dopamine, significantly contributes to the neurodegenerative processes linked to PD. In a variety of dopaminergic cell models, such as PC12 and N27 mesencephalic cells, NAC has consistently shown protective effects against dopamine-induced injury. In 2011, Jana et al. examined the mechanisms underlying dopamine-induced toxicity in both isolated rat brain mitochondria and PC12 neuronal cells, and clarified the protective function of NAC [85]. Dopamine induced membrane depolarization and a reduction in phosphorylation capacity in isolated mitochondria in a dose-dependent manner. Oxyradical scavengers or metal chelators did not mitigate this damage. However, it was significantly inhibited by reduced GSH and NAC, and exacerbated by tyrosinase, strongly suggesting that the quinone oxidation products of dopamine were the main agents responsible for mitochondrial dysfunction. In PC12 cells, exposure to dopamine resulted in a marked disruption of mitochondrial bioenergetic functions, leading to an approximately 40% decrease in cell viability, along with signs of apoptotic nuclear alterations and heightened activities of caspase 3 and caspase 9. NAC successfully mitigated all cytotoxic effects caused by dopamine, actively suppressing the increase in ROS production, the formation of quinoprotein adducts within mitochondria, and the accumulation of quinone products in the culture medium [85].

Additionally, in PC12 cells, NAC reliably safeguarded cell viability against dopamine or dopamine in conjunction with Z-ligustilide or 1-methyl-4-phenylpyridinium [86,87]. This protective effect was frequently associated with the restoration of intracellular GSH levels and the reduction in dopamine quinone/melanin formation, highlighting NAC’s capacity to neutralize dopamine-derived toxic substances. Importantly, non-thiol antioxidants typically exhibited limited or no efficacy, underscoring NAC’s distinctive thiol-based mechanism [86,88].

Mechanistically, NAC’s protective effects against dopamine toxicity also included its impact on mitochondrial function. It was observed that NAC could reverse the inhibition of mitochondrial complex I/IV that dopamine induced in PC12 cells [84]. Furthermore, in N27 mesencephalic cells, NAC demonstrated the ability to prevent apoptosis and proteasome inhibition triggered by dopamine, suggesting that it mitigates oxidative stress associated with dopamine rather than directly inhibiting the proteasome [89]. Shivalingappa et al. (2012) highlighted the importance of redox homeostasis for the survival of dopaminergic cells, demonstrating that NAC alleviates methamphetamine-induced neurotoxicity in rat brain dopaminergic neurons by restoring redox balance, reducing oxidative stress, and partially inhibiting autophagy and apoptosis [90].

Collectively, these cell culture investigations underscore that dopamine toxicity primarily disrupts redox balance and mitochondrial function, particularly in dopaminergic cells. NAC effectively counteracts these effects by replenishing GSH, scavenging dopamine-derived quinones, and safeguarding the enzymes of the electron transport chain. These mechanisms contribute to the preservation of dopaminergic neuron integrity and functionality, even in the presence of oxidative stress conditions.

Redox-based protection in various in vivo models

Berman et al. (2011) investigated age-related degeneration in EAAC1(-/-) mice, a PD model with impaired cysteine transport and sustained GSH depletion [45]. The mice exhibited progressive loss of dopaminergic neurons in the SN *pars compacta*, accompanied by nitrosative stress markers (nitrotyrosine, nitrosylated α-synuclein) and microglial activation. NAC administration significantly attenuated these effects, suggesting that its ability to replenish GSH may mitigate age-dependent dopaminergic decline.

Expanding on these discoveries, numerous investigations utilizing the 1-methyl-4-phenyl-1,2,3,6-tetrahydropyridine (MPTP) mouse model—a well-known neurotoxin that specifically affects nigrostriatal dopaminergic neurons—consistently demonstrate significant neuroprotective benefits of NAC. According to He et al. (2015) the oral 14-day administration of NAC improved motor performance, preserved tyrosine hydroxylase (TH)-positive neurons, and reversed MPTP-induced reductions in dopamine, 3,4-Dihydroxyphenylacetic acid (DOPAC), and homovanillic acid [91]. Mechanistically, these effects were associated with NAC’s capacity to inhibit mitochondrial apoptotic signaling (which includes the release of cytochrome c and the activation of caspases 3/6/9), diminish ROS accumulation, and inhibit the phosphorylation of JNK and p38. Previous MPTP research corroborates these findings: NAC pretreatment alleviated the depletion of dopamine and DOPAC, reversed the loss of GSH, and reinstated the functionality of antioxidant enzymes including GPx and SOD [92,93]. Additionally, NAC exhibited anti-inflammatory actions by reducing IL-6 and TNF-α expression at both mRNA and protein levels and by enhancing GPx gene transcription.

Comparable neuroprotective mechanisms have been identified in the rotenone model, which causes inhibition of mitochondrial complex I and replicates various features associated with PD. Rahimmi et al. (2015) demonstrated that intraperitoneal administration of NAC improved motor coordination, reversed dopamine loss in the SN, and normalized the levels of essential mitochondrial proteins—decreasing parkin and increasing dynamin-related protein 1 (Drp1)—thus indicating a protective effect against mitochondrial fragmentation and disrupted turnover [94].

In a complementary rat model that involved both paraquat and zinc, Kumar et al. (2012) illustrated that NAC effectively prevented the loss of striatal dopamine and TH, improved behavioral deficits, and diminished lipid peroxidation along with NADPH oxidase activity [95]. Additionally, NAC reduced apoptotic signaling (including cytochrome c release and caspases 3/9) and neuroinflammation (as indicated by CD11b expression). The concurrent administration of apocyanin, which acts as an NADPH oxidase inhibitor, further enhanced these protective effects, thereby underscoring the significant role of ROS-mediated apoptosis in the degeneration of dopaminergic neurons.

Together, these preclinical models consistently demonstrate that NAC exerts neuroprotective effects by counteracting mitochondrial dysfunction, restoring redox balance, suppressing inflammation, and preserving dopaminergic integrity.

Modulation of Dopamine Transporter Activity

NAC affects the levels and activity of dopamine transporter (DAT) in the brain through context-dependent mechanisms, contributing to its therapeutic potential in dopaminergic disorders such as PD.

Preclinical studies suggest that NAC exerts bidirectional effects on DAT, modulating its expression according to the prevailing dopaminergic state. In a hemiparkinsonian rat model, treatment with NAC led to a reduction in DAT levels within the striatum of the non-lesioned hemisphere, which is understood as a compensatory response aimed at balancing dopamine levels [14]. Furthermore, in tackling the ongoing issue of motor impairment and disease progression associated with PD, Caridade-Silva et al. (2023) investigated whether NAC could alleviate these challenges by modulating dopaminergic transmission [15]. Utilizing a rat model of PD induced by 6-hydroxydopamine (6-OHDA), NAC was found to significantly enhance the survival of dopaminergic neurons and restore DAT levels to normal. Significantly, the histological enhancements noted were directly associated with a marked recovery in motor function among the affected animals. This study provides compelling evidence indicating that NAC could possess therapeutic potential in addressing the degenerative mechanisms linked to PD. This suggests that NAC may have dual effects on DAT levels, depending on the pathophysiological circumstances, potentially restoring dopaminergic transmission in both surplus and deficit conditions. Recent investigations using an in vitro model of PD have additionally demonstrated that NAC treatment can alleviate the detrimental effects of oxidative stress on DAT expression and functionality [16]. By sustaining appropriate DAT levels and activity, NAC contributes to the preservation of the spatial and temporal regulation of dopaminergic signaling, which is vital for normal motor function and reward processing.

To investigate the clinical and biological effects of NAC on PD due to a lack of existing data, Monti et al. (2016, 2019) performed a randomized trial [96,97]. Forty-two PD patients were assigned to receive either a combination of weekly intravenous and daily oral NAC or standard care, and exhibited a significant increase in DAT binding in the caudate and putamen regions, which was associated with substantial improvements in PD symptoms. Additionally, a significant secondary finding was the increased binding of midbrain serotonin transporter (SERT). These findings indicate that NAC positively affects the dopaminergic system in PD and shows potential as a therapeutic option, necessitating further large-scale controlled studies.

Influence on Dopamine Metabolism and Turnover

NAC influences several components of dopaminergic signaling, including synthesis, storage, release, reuptake, and protection against oxidative stress.

Gere-Pászti and Jakus (2009), who employed an in vitro striatal slice model alongside a double-amphetamine (AMPH) challenge to evaluate DA efflux, presented initial findings regarding the influence of NAC on dopamine release [13]. High concentrations of NAC (10 mM) entirely suppressed AMPH-induced dopamine release, and additional data obtained through reserpine indicated that this phenomenon was linked to the subcellular positioning of vesicular dopamine, suggesting a direct modulatory effect on atypical DA efflux. In vivo support for this mechanism came from Bauzo et al. (2012), who demonstrated that intramuscular NAC administration in squirrel monkeys attenuated cocaine-induced elevations in extracellular dopamine, while leaving basal levels unchanged [98]. These findings are consistent with NAC’s ability to stabilize glutamatergic transmission, which indirectly shapes dopaminergic tone via cystine–glutamate exchange.

Beyond release modulation, NAC exerts protective effects on dopamine systems by mitigating oxidative stress. In a rodent model of amphetamine-induced neurotoxicity, NAC reversed striatal dopamine depletion and significantly reduced oxidative damage markers, such as lipid peroxidation and hydroxyl radical production [99]. More recent findings by El-Habta et al. (2024) in in vitro models simulating dopaminergic neurodegeneration confirmed that NAC protects against oxidative impairment of both dopamine levels and transporter function, reinforcing its neuroprotective role under conditions of redox imbalance [16].

In addition to reducing dopamine loss, NAC appears to support dopamine synthesis and storage. Recent research findings in a rat model of PD indicate that NAC administration preserves the activity of tyrosine hydroxylase (TH) and enhances the expression of vesicular monoamine transporter 2 (VMAT2), thus ensuring the stability of dopamine production and storage [15,16]. Consistent with these findings, El-Habta et al. (2024) showed that NAC treatment elevated total dopamine levels in cellular models of dopaminergic degeneration, suggesting improved neurotransmitter homeostasis [16].

NAC’s impact on dopamine metabolism also encompasses its function in the context of addictive behaviors. Laverde et al. (2021) demonstrated that NAC blocked the development of ethanol-induced conditioned place preference in mice, indicating a disruption of reward-related associative learning [100]. On a neurochemical level, NAC enhanced dopamine turnover in the SN and increased dopamine levels in the NAc of ethanol-treated animals. These effects suggest that NAC can suppress the development of alcohol-related reinforcement while maintaining normal dopaminergic transmission.

Together, these findings support a multifaceted role for NAC in dopaminergic regulation, encompassing both direct and indirect pathways, could play a role in its therapeutic efficacy in substance use disorders and other conditions associated with dysregulated reward pathways [101].

Indirect Modulation via Glutamatergic, Neuroplastic, and Neuro-Immune Pathways

In addition to its influence on dopamine synthesis and transport, NAC also indirectly regulates dopaminergic function by affecting glutamatergic signaling, neuronal plasticity, and neuroimmune interactions [101]. As mentioned in Section 3.2.1, Baker et al. (2002) showed that dopaminergic transmission is affected by its interaction with the extracellular glutamate that is released by the cystine/glutamate antiporter (xCT) [73]. Building on these foundational insights, Lai et al. (2022) provided compelling evidence of NAC’s multifaceted neuromodulatory capacity in a Disc1 mutant mouse model of prodromal schizophrenia [102]. NAC treatment normalized the imbalance between striatal dopamine D1 and D2 receptors and restored glutamatergic function by repairing disrupted neuronal morphology. Notably, NAC elevated striatal levels of glycogen synthase kinase 3 (GSK3), a key regulator of dendritic structure, which led to significant recovery of dendritic complexity and spine density. The study also highlighted NAC’s impact on neuroimmune regulation by improving microglial morphology and neuron–microglia interactions, mechanisms essential for maintaining synaptic integrity and preventing inflammation-driven neuronal damage. Together, these effects contributed to NAC’s protective action against psychosis-like behaviors in the Disc1 model.

In summary, preclinical research consistently demonstrates NAC’s multifaceted modulation of dopaminergic system, signifying its therapeutic potential across diverse neurological pathologies. In models of PD and dopamine toxicity, NAC robustly protects dopaminergic neurons by restoring redox homeostasis, enhancing mitochondrial integrity, mitigating oxidative stress, and countering neuroinflammation and apoptosis.

#### 3.4.3. GABAergic System

Although NAC does not act directly on the GABAergic system, its effects on glutamate regulation, oxidative stress, and astrocyte function indirectly support inhibitory neurotransmission. Several lines of evidence suggest that NAC preserves GABAergic tone by modulating upstream glutamatergic pathways, restoring astrocytic glutamate transport, and mitigating oxidative damage, especially in the context of neurotoxic and neuropsychiatric conditions.

In vitro studies have shown NAC’s protective effects against such imbalances. For example, in cultured rat striatum and ventral mesencephalon, a 24 h exposure to polychlorinated biphenyls resulted in a decrease in both dopamine and GABA levels [103]. Pretreatment with NAC completely restored the GABA depletion and alleviated the reductions in dopamine and GABA levels caused by hydrogen peroxide. These findings underscore NAC’s neuroprotective role in preserving inhibitory tone under redox imbalance.

Astrocytic Glutamate Transport and Heterosynaptic Regulation

A primary mechanism by which NAC supports GABAergic function is through restoration of glutamate balance, essential for maintaining inhibitory signaling. Astrocytes serve as essential regulators of extracellular glutamate levels and preserving synaptic homeostasis. NAC maintains the excitatory–inhibitory equilibrium by upregulating GLT-1 (also referred to as the excitatory amino acid transporter, EAAT2), thereby enhancing glutamate uptake and preventing toxic spillover. Roberts-Wolfe and Kalivas (2015) emphasized the therapeutic relevance of this mechanism, particularly in addiction [76]. In cocaine self-administration models, NAC restored reduced GLT-1 expression in the NAc and attenuated relapse-like drug-seeking behaviors [104]. The specific mechanisms that underlie NAC’s ability to enhance EAAT2 expression have yet to be completely clarified, but they are believed to involve the transcriptional regulation of the EAAT2 gene [5,105,106]. Comparable positive effects on glutamate homeostasis via EAAT2 modulation have been shown in experimental models of stroke and various neurodegenerative disorders, underscoring the potential of NAC to restore glutamate equilibrium [5]. Durieux et al. (2015) highlighted the role of astrocytic xCT modulation by NAC in restoring striatal glutamate levels and the broader excitatory/inhibitory balance relevant to neurodevelopmental disorders such as autism and schizophrenia [107]. They showed that NAC administration reduced striatal glutamate in mice, indirectly influencing GABAergic tone through excitatory pathway regulation. Lau et al. (2021) further demonstrated that NAC counters obesity-induced astrocyte dysfunction in rats, which otherwise leads to impaired glutamate clearance and disrupted GABA transmission in the orbitofrontal cortex [79]. By restoring GLT-1 activity, NAC preserved heterosynaptic regulation and protected inhibitory signaling.

GABA Levels and Receptor Interaction: Limited and Conflicting Evidence

Although NAC indirectly influences GABAergic neurotransmission through its modulation of glutamatergic tone and astrocytic function, direct effects on GABA levels remain inconclusive. A double-blind, placebo-controlled clinical trial by Schulte et al. (2017) investigated the impact of NAC on glutamate and GABA concentrations in the dorsal anterior cingulate cortex of smokers [108]. The results showed no significant changes in either neurotransmitter, suggesting that NAC’s neuromodulatory effects may be region-specific and dependent on population characteristics and underlying pathophysiology. In addition, Beltrán González et al. (2018) reported in vitro evidence that L-cysteine, a downstream metabolite of NAC, may act as a negative allosteric modulator of GABAAρ1 receptors [109]. However, this effect has not yet been investigated in vivo, and its physiological significance in the brain remains ambiguous.

#### 3.4.4. Serotonergic System

NAC exerts multifaceted effects on serotonergic neurotransmission, largely mediated through its antioxidant properties, modulation of glutamatergic pathways, anti-inflammatory actions, and regulation of serotonin levels and receptor function [110]. These mechanisms collectively support NAC’s therapeutic potential in conditions involving serotonergic dysfunction, such as depression, anxiety, neurotoxicity, and psychosis.

Serotonin Preservation through Antioxidant and Anti-inflammatory Action

NAC’s significant antioxidant properties are crucial for mitigating the impact of oxidative stress in CNS [111]. Chopra et al. (2021) investigated the effects of NAC in a rodent model of manganese-induced neurotoxicity, which produced behavioral deficits, elevated oxidative stress markers, and significantly reduced 5-HT levels in the cortex and midbrain [110]. NAC administration mitigated these effects by restoring redox balance, improving acetylcholinesterase activity, reducing behavioral impairments, and significantly normalizing midbrain serotonin levels. These findings underscore NAC’s capacity to protect serotonergic neurons from oxidative damage. Similarly, in a model examining neurotoxicity induced by propionic acid, Aldbass et al. (2023) illustrated that NAC mitigated serotonin depletion, decreasing a 33% reduction to 14% [112]. In addition to enhancing serotonin levels, NAC attenuated neuroinflammatory markers (e.g., IFN-γ), increased GST activity, and reduced DNA damage, highlighting its broad neuroprotective effects in toxicological contexts.

Comparable protective effects are detected in chronic stress models. Fernandes and Gupta (2019) demonstrated that NAC mitigated depressive-like behaviors and reinstated hippocampal and cortical serotonin levels in rats subjected to chronic unpredictable mild stress, while also decreasing heightened pro-inflammatory cytokines (IL-1β, IL-6, TNF-α) [113]. These findings were consistent with the effects of fluoxetine, reinforcing the significance of NAC in modulating both inflammatory and serotonergic mechanisms in stress-related conditions.

Modulation of Serotonin Levels and Receptor Function

NAC has been demonstrated to have a direct impact on serotonin levels and receptor signaling, frequently in association with glutamatergic modulation. In models of depression induced by alcohol withdrawal, Yawalkar et al. (2018) found that NAC elevated serotonin levels in the hippocampus and ameliorated behaviors characteristic of depression [114]. These outcomes were linked to the normalization of NMDA receptor signaling, highlighting the interconnectedness of the glutamatergic and serotonergic systems.

Further insights come from receptor-level investigations. In 2014 Lee et al. reported that NAC attenuated behavioral and molecular responses induced by the hallucinogenic 5-HT2A receptor agonist, (±)1-(2,5-dimethoxy-4-iodophenyl)-2-aminopropane (DOI) in mice [115]. This encompassed decreases in head twitch responses and immediate early gene expression (c-Fos, Egr-2) within the cortex, along with the suppression of DOI-induced excitatory field potentials. The fundamental mechanism was characterized by an increased function of the xCT transporter, which resulted in the activation of metabotropic glutamate receptor 2 (mGluR2). The pharmacological inhibition of xCT or mGluR2 negated the effects of NAC, thereby affirming the significance of this interaction between glutamate and serotonin receptors. 5-HT_2A_ receptors and mGluR2 are both found in cortical pyramidal neurons and may create a 5-HT_2A_-mGluR2 complex that is responsible for distinct cellular responses when engaged by hallucinogenic substances. By modulating this complex, NAC could offer therapeutic benefits in conditions marked by dysregulated 5-HT_2A_ receptor function, such as schizophrenia and hallucinogen use disorders [115].

Additionally, a clinical case report by Lafleur et al. (2006) showed that NAC supplementation improved symptoms in a patient with obsessive–compulsive disorder unresponsive to serotonin reuptake inhibitors, further reinforcing the translational potential of these mechanisms [116].

#### 3.4.5. Adrenergic System

Although less extensively studied, NAC appears to influence adrenergic neurotransmission primarily through its antioxidant properties. Oxidative stress is known to impair nitric oxide (NO) bioavailability, which in turn modulates norepinephrine (NE) activity. NAC preserves NO from oxidation, indirectly regulating NE overflow and adrenergic tone [10,16,101,117,118]. This modulation has an inhibitory impact on NE-induced vasoconstriction [117]. In situations characterized by induced oxidative stress, particularly in hypertensive rat models, there is an augmented release of NE resulting from reduced bioavailability of NO. While the direct effects on the CNS have yet to be completely clarified, this mechanism could play a significant role in the neurovascular dysregulation associated with stress.

#### 3.4.6. Cholinergic System

NAC modulates cholinergic neurotransmission through a dual mechanism: (1) restoring enzymatic homeostasis critical to acetylcholine (ACh) metabolism, and (2) protecting cholinergic neurons from oxidative stress and neurotoxic insults that impair cognitive function.

Restoration of Enzymatic Homeostasis in Acetylcholine Metabolism

NAC has been shown to normalize the activities of key enzymes involved in ACh metabolism. In mouse models of cognitive dysfunction induced by streptozotocin, NAC inhibited the pathological increase in acetylcholinesterase (AChE) activity in the brain cortex and hippocampus, as well as butyrylcholinesterase (BChE) activity in the hippocampus [119].

Additionally, NAC mitigates reductions in choline acetyltransferase (ChAT) activity under oxidative stress conditions. In a rat model simulating tyrosinemia type II through chronic L-tyrosine administration, NAC co-treatment effectively prevented the decline in ChAT activity in the cerebral cortex and partially reversed the increase in AChE activity in the hippocampus and striatum [120].

While these enzymatic changes were not consistently evident in noticeable memory deficits, they indicate that NAC preserves cholinergic function by facilitating both the synthesis and breakdown pathways of ACh.

Protection against Neurotoxicity and Metabolic Support

NAC also provides significant neuroprotection to cholinergic neurons by mitigating oxidative damage and counteracting various neurotoxins. For instance, in a study examining cadmium-induced neurotoxicity in rats, NAC prevented the associated cognitive impairments, restored AChE activity, and reduced lipid peroxidation in the hippocampus, cerebellum, and hypothalamus [121]. Similar protective effects were observed in SN56 cholinergic cells exposed to the neonicotinoid biocide imidacloprid, where NAC diminished oxidative stress and preserved cholinergic integrity [122]. The co-administration of NAC further reinforces its neuroprotective and redox-modulating properties in chemically induced neurotoxicity models, as shown by its ability to reduce CIS-induced increases in AChE and monoamine oxidase activities in a rodent model [33].

Together, these effects maintain the integrity of cholinergic neurons, thus facilitating cognitive functions and alleviating the impacts of neurodegenerative disorders [121,122].

In addition to its role in enzyme regulation and oxidative defense, NAC also indirectly enhances cholinergic function by preserving metabolic processes. It specifically mitigates the reduction in hippocampal glucose uptake that results from neurotoxic damage [119]. The production of acetylcholine (ACh) is dependent on acetyl-CoA, which is generated from glucose metabolism [119]. Consequently, the glucose-preserving effects of NAC indirectly facilitate the synthesis of acetylcholine and contribute to the overall integrity of cholinergic systems.

#### 3.4.7. Glutamine Synthesis

As previously noted, NAC is widely recognized for its capacity to boost intracellular GSH synthesis [123]. Acting as a cysteine donor, NAC provides the rate-limiting substrate for GSH production. Importantly, this redox-supportive role intersects with the glutamate–glutamine cycle, a central regulator of neurotransmitter balance and oxidative homeostasis in the central nervous system [124]. Disruption of this cycle may impair GSH biosynthesis and exacerbate excitotoxic damage, particularly under neuroinflammatory conditions.

NAC enhances the function of glutamine synthase (GS), which facilitates the transformation of glutamate into glutamine and maintains neurotransmitter balance. Visalli et al. (2007) demonstrated that exposing human astrocytes to the HIV-1 envelope glycoprotein gp120 led to significant neurotoxic consequences, including diminished cell viability, increased apoptosis, lipid peroxidation, and, reduced expression of GS [125]. Pre-treatment with NAC mitigated these adverse effects in a dose-dependent manner, with elevated NAC concentrations (up to 5 mM) offering greater protective benefits. Western blot analysis and immunostaining demonstrated that NAC restored GS protein levels, while biochemical assays confirmed the recovery of glutamine concentrations in the culture media of astrocytes. Moreover, markers of oxidative damage, including MDA, were reduced in cells that received NAC treatment. Mechanistically, NAC’s protective role appears dual: it attenuates gp120-induced oxidative stress through its antioxidant properties, and it reinstates GS function, promoting glutamate detoxification and supporting neurotransmitter cycling. Given that gp120 impairs glutamatergic neurotransmission by suppressing GS leading to extracellular glutamate accumulation and glutamine deficiency. NAC’s ability to counteract this imbalance may mitigate excitotoxicity and neurodegeneration [125].

#### 3.4.8. Cortisol Transmission

NAC exerts context-dependent effects on the hypothalamic–pituitary–adrenal (HPA) axis, modulating both glucocorticoid secretion and receptor expression. Evidence suggests a bidirectional influence, with NAC promoting hyperactivity in otherwise healthy HPA systems, while offering restorative effects in conditions of stress-induced dysregulation. In a study by Prevatto et al. (2017), combined administration of NAC and vitamin E in Wistar rats resulted in elevated plasma corticosterone levels, indicative of HPA axis hyperactivation [126]. This effect was accompanied by increased expression of melanocortin receptor type 2 (MC2R) and steroidogenic acute regulatory protein (StAR) in the adrenal glands, alongside decreased glucocorticoid (GR) and mineralocorticoid (MR) receptor expression in the pituitary. These findings suggest that NAC-induced reduction in oxidative stress may disrupt redox-sensitive regulatory mechanisms within the HPA axis, impairing the negative feedback loop and promoting adrenal stimulation. Furthermore, NAC was found to inhibit dexamethasone-induced suppression of corticosterone, reinforcing its role in blunting glucocorticoid-mediated feedback inhibition.

Chaves et al. (2025) reported complementary findings, noting that NAC reduced the expression of Nrf2 in the pituitary gland, which resulted in adrenal hypertrophy and hypercorticoidism [127]. This shift was associated with a reduction in GRα and a rise in GRβ expression—two isoforms of the GR that influence feedback sensitivity. Importantly, activation of the Nrf2–heme oxygenase-1 (HO-1) pathway via cobalt protoporphyrin IX reversed changes, normalizing corticosterone levels and receptor expression. These findings suggest that NAC may exert its effects on HPA axis regulation through redox-sensitive pathways, particularly via Nrf2–HO-1 signaling [127].

In contrast, under conditions of stress-related dysfunction of the HPA axis, NAC seems to exert a normalizing influence. Prolonged stress and inflammation can impair the functioning of the HPA axis, leading to altered cortisol secretion [128]. Research conducted by Korou et al. (2014) demonstrated that middle-aged mice on a high-cholesterol diet showed an increase in GR expression within the hypothalamus—a disruption linked to HPA axis dysregulation [129]. The administration of NAC inhibited this upregulation, underscoring its potential role in safeguarding against neuroendocrine dysfunction induced by metabolic factors. Chakraborty et al. (2020) further demonstrated that chronic NAC treatment in a rat model of clomipramine-induced anxiety and depression normalized elevated corticosterone levels, reversed noradrenergic dysfunction in the amygdala, and reduced hypertrophy of the adrenal glands and spleen [130]. These effects suggest that NAC can ameliorate stress-induced alterations in HPA activity, potentially through anti-inflammatory and neuromodulatory mechanisms.

The neuromodulatory properties of NAC are schematically presented in Figure 3.

### 3.5. Modulation of Gene Expression

NAC modulates the expression of genes involved in neurotransmission and stress adaptation, contributing to its neuroprotective and antidepressant properties.

Yawalkar et al. (2018) demonstrated that NAC reversed alterations in NMDA receptor subunit gene expression associated with ethanol abstinence [114]. In their study, rats exposed to prolonged voluntary alcohol consumption followed by withdrawal exhibited significant upregulation of GRIN2A and GRIN2B (encoding NMDA receptor subunits) in both the hippocampus and prefrontal cortex, along with depression-like behaviors. NAC administration during the abstinence phase significantly reduced immobility times in behavioral tests and normalized GRIN2A and GRIN2B expression. Additionally, NAC restored hippocampal and plasma serotonin levels at both doses, with the higher dose (100 mg/kg) also increasing serotonin in the prefrontal cortex [114]. In a separate study, Brivio et al. (2023) explored NAC’s ability to regulate immediate early gene (IEG) expression in response to acute stress [7]. Adult male rats underwent a treatment regimen lasting 21 days with either venlafaxine or NAC, followed by exposure to acute restraint stress. NAC demonstrated superior efficacy compared to venlafaxine in enhancing IEG expression within the ventral hippocampus and amygdala, which are critical regions involved in the regulation of emotion and stress. Importantly, pretreatment with NAC resulted in a significant upregulation of Nr4a1 and c-Fos in the ventral hippocampus, as well as Nr4a1, c-Fos, and Arc in the amygdala. These genes are recognized as indicators of neuronal activity and plasticity. Notably, Nr4a1 is rapidly activated in response to stress and plays essential roles in synaptic remodeling, mitochondrial function, and the regulation of cellular energy. Furthermore, the authors pointed out a possible connection between Nr4a1 and Gpx1 transcription, indicating that the antioxidant properties of NAC may be partially mediated through the modulation of stress-responsive transcription factors.

Collectively, these results suggest that NAC not only mitigates maladaptive gene expression patterns associated with substance withdrawal and stress but also prepares neural circuits for adaptive plasticity. Its capacity to influence both neurotransmission-related and plasticity-related genes establishes NAC as a potential agent for improving neurobiological resilience.

## 4. Discussion and Future Perspectives

This review synthesizes the expanding collection of evidence that underscores N-acetylcysteine as a versatile neuromodulatory agent with considerable implications for the management of CNS disorders. The therapeutic adaptability of NAC is founded on its strong antioxidant properties, primarily through the restoration of intracellular glutathione, which mitigates oxidative stress—a common pathological characteristic in neurodegenerative and neuropsychiatric conditions. Importantly, NAC’s effects go beyond maintaining redox equilibrium to encompass the modulation of neuroinflammatory processes, neurotransmitter systems, mitochondrial functionality, and neurotrophic support.

Mechanistic investigations reveal that NAC produces both glutathione-dependent and—independent effects. Its indirect stimulation of antioxidant defenses via the Nrf2-ARE pathway and the regulation of mitochondrial sulfane sulfur metabolism establish NAC as a promising candidate for conditions characterized by oxidative and mitochondrial dysfunction. Furthermore, the compound reduces neuroinflammation by inhibiting critical mediators such as NF-κB, iNOS, and pro-inflammatory cytokines, while also modulating microglial signaling. These effects are particularly pertinent in disorders where neuroinflammation is a central etiological factor, such as AD and PD.

Furthermore, NAC exhibits significant effects across various neurotransmitter systems, particularly within the glutamatergic and dopaminergic pathways. It promotes the clearance of glutamate by upregulating GLT-1 and regulates synaptic glutamate release through the cystine–glutamate antiporter system, thereby aiding in the maintenance of excitatory-inhibitory balance and providing neuroprotection. This mechanism may explain its potential utility in addressing conditions such as addiction, schizophrenia, and mood disorders. In a similar vein, NAC’s impact on dopaminergic transmission—a through redox regulation, modulation of dopamine transporters and effects on dopamine metabolism—highlights its importance in the context of PD and substance use disorders. While NAC is not a direct GABAergic agent, its indirect actions, particularly through restoration of glutamate homeostasis, upregulation of astrocytic GLT-1, and modulation of excitatory-inhibitory balance—create a neural environment supportive of inhibitory tone. The evidence points to an indirect yet potentially clinically significant role of NAC in maintaining inhibitory tone, particularly in situations involving neuroinflammation, excitotoxicity, or synaptic imbalance. However, additional research is required to elucidate its impact on GABA synthesis, receptor function, and interneuron modulation.

Despite strong preclinical evidence, clinical results have been variable. Meta-analyses and clinical trials have produced inconsistent findings, especially within addiction and psychiatric populations, which may be attributed to variations in dosing strategies, patient selection criteria, and outcome assessment methods. These discrepancies emphasize the necessity for standardized clinical protocols and biomarker-driven strategies to identify sub-groups that are likely to respond favorably.

Besides its impact on classical neurotransmitters, NAC seems to affect the expression of neurotrophic factors, epigenetic regulation, and endocrine signaling also (for instance, modulation of the HPA axis). These extensive neuromodulatory effects indicate its potential as a supplementary treatment for intricate, multifactorial CNS disorders. Nevertheless, challenges such as low oral bioavailability and restricted CNS penetration necessitate further optimization of pharmacokinetics and the development of formulations.

Given its extensive range of mechanisms, N-acetylcysteine offers a distinctive opportunity to tackle the complex nature of CNS disorders. Nevertheless, several significant gaps must be addressed in forthcoming research. Firstly, enhancing NAC’s bioavailability and CNS penetration through improved delivery methods is crucial. Innovative formulations, such as liposomal, intranasal, or prodrug derivatives, could potentially mitigate the challenges associated with first-pass metabolism and restricted blood–brain barrier permeability. Secondly, there is a pressing need for a stronger focus on translational research to connect promising preclinical findings with inconsistent clinical outcomes. This necessitates the utilization of well-defined patient cohorts, standardized outcome measures, and validated biomarkers to pinpoint individuals who are most likely to gain from NAC therapy.

Moreover, future investigations should explore the synergistic effects of NAC in conjunction with other pharmacological or behavioral interventions, particularly in psychiatric and neurodegenerative settings where multimodal treatment approaches are frequently required. The examination of NAC’s epigenetic, neuroendocrine, and immunomodulatory characteristics may also unveil new therapeutic targets. Long-term safety assessments and dose-optimization studies are essential to refine treatment protocols, particularly for chronic administration in at-risk populations. Lastly, broadening the research into less-explored neurotransmitter systems and neurodevelopmental disorders could reveal additional clinical applications.

## 5. Critical Interpretation

Accumulating evidence underscores the diverse neuromodulatory effects of N-acetylcysteine (NAC), with substantial preclinical findings affirming its role in regulating oxidative stress, neuroinflammation, neurotransmission, and neuronal viability. However, a significant translational gap remains between experimental models and clinical results. This discrepancy necessitates a thorough critical analysis of the existing literature.

A major obstacle is the limited bioavailability of NAC, especially after oral intake, due to significant first-pass hepatic metabolism. While certain studies indicate elevated central glutathione levels with intravenous or high-dose oral NAC, others do not replicate these results, particularly in clinically relevant brain areas such as the SN. This inconsistency prompts inquiries into the pharmacokinetic thresholds required to elicit central therapeutic effects.

Furthermore, clinical trials frequently exhibit variability in their design, encompassing differences in treatment duration, dosing regimens (ranging from 600 to 6000 mg/day), methods of administration, and the endpoints evaluated. In numerous randomized controlled trials, especially those focused on substance use disorders or psychiatric conditions, NAC has not consistently shown superiority over placebo. These variations may arise from the absence of biomarker-guided stratification or patient subgrouping based on redox status, glutamate dysregulation, or inflammatory profiles, which could enhance understanding of treatment responsiveness.

The multimodal characteristics of NAC’s mechanisms, although promising, also bring about interpretive intricacies. For example, the concurrent modulation of glutamatergic and dopaminergic pathways requires a nuanced neurochemical equilibrium, which may differ significantly across various neuropsychiatric conditions and stages of disease. Moreover, recent findings indicate that elevated doses might lead to cytotoxic effects, especially within microglial populations, underscoring the necessity for meticulously adjusted dosing protocols.

Additionally, numerous studies overlook the long-term consequences or the duration of NAC’s neurobiological impacts following acute administration. Considering its ability to affect gene expression, mitochondrial function, and synaptic plasticity, longitudinal studies are essential for assessing the sustained therapeutic significance.

In summary, while NAC shows substantial promise as an adjunctive or repurposed treatment for a range of CNS disorders, its clinical effectiveness may hinge on precision-medicine strategies—such as optimized formulations, biomarker-guided patient selection, and comprehensive therapeutic approaches. Future research should emphasize mechanism-oriented designs, thorough pharmacodynamic assessments, and stratified patient cohorts to more clearly define NAC’s role in neurotherapeutics.

## 6. Conclusions

NAC emerges from this review as a promising, well-tolerated candidate for neurotherapeutic applications. Its pleiotropic mechanisms—spanning antioxidant, anti-inflammatory, neuromodulatory, and gene-regulatory pathways—support its use in a range of neurodegenerative and psychiatric disorders. Future research should prioritize high-quality and mechanistically informed clinical trials, and should explore novel delivery systems to maximize NAC’s therapeutic potential in the CNS.

## Figures and Tables

**Figure 1 cimb-47-00710-f001:**
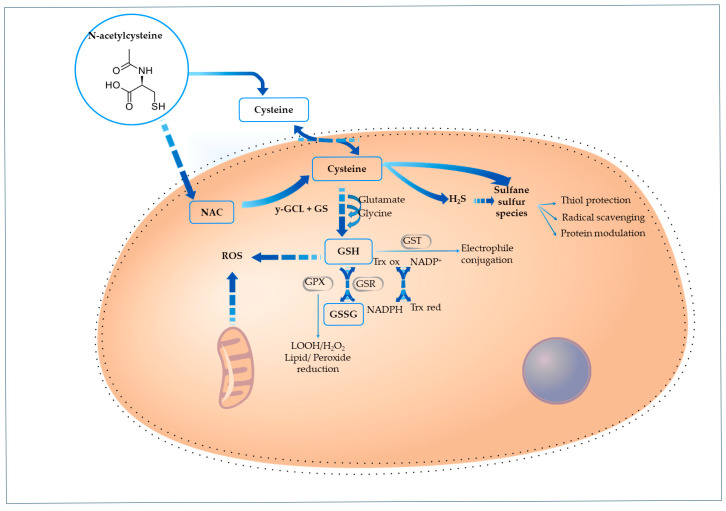
Cellular pathways through which N-acetylcysteine supports redox balance and sulfur-based signaling. NAC is absorbed by cells and converted to cysteine through deacetylation, which plays a role in two primary pathways: glutathione synthesis and sulfur species formation. Cysteine is converted to glutathione (GSH) through the sequential action of γ-glutamylcysteine ligase (γ-GCL) and glutathione synthetase (GS). GSH acts as a key antioxidant by neutralizing reactive oxygen species (ROS) directly and via enzymatic systems: glutathione peroxidase (GPX), glutathione S-transferase (GST), which mediate peroxide reduction, electrophile detoxification, and disulfide bond regulation, respectively. Cysteine is also a precursor for hydrogen sulfide (H_2_S) and other sulfane sulfur species, which provide additional protective roles through radical scavenging, thiol group preservation, and redox-sensitive protein modulation.

**Figure 2 cimb-47-00710-f002:**
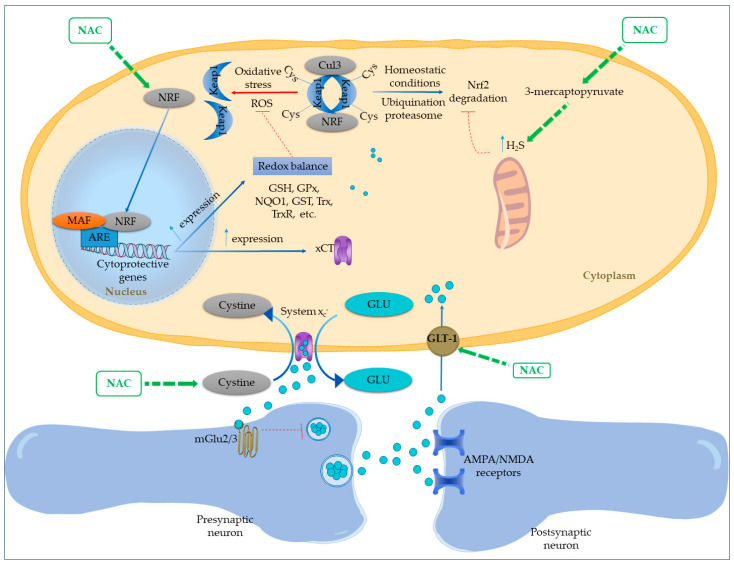
Proposed mechanisms of N-acetylcysteine action on redox regulation and glutamatergic signaling via Nrf2–ARE pathway activation. NAC enhances nuclear translocation of Nrf2 by disrupting Keap1-mediated degradation, either directly under oxidative stress or indirectly through increased mitochondrial hydrogen sulfide (H_2_S) production from 3-mercaptopyruvate metabolism. H_2_S modifies cysteine residues on Keap1, impairing its repressive function and releasing Nrf2. Once in the nucleus, Nrf2 activates antioxidant response element (ARE)-driven gene transcription, upregulating key antioxidant systems including GSH, GPx, NQO1, GSTs, Trx, and TrxR. This contributes to redox homeostasis and may also enhance NADPH production through the pentose phosphate pathway. NAC further promotes cystine uptake via xCT (System x^−^), facilitating both GSH biosynthesis and glutamate efflux. The extracellular glutamate stimulates presynaptic inhibitory mGlu2/3 receptors, reducing glutamate release and excitotoxicity. In parallel, NAC increases astrocytic GLT-1 expression, supporting glutamate clearance and maintaining synaptic stability.

**Figure 3 cimb-47-00710-f003:**
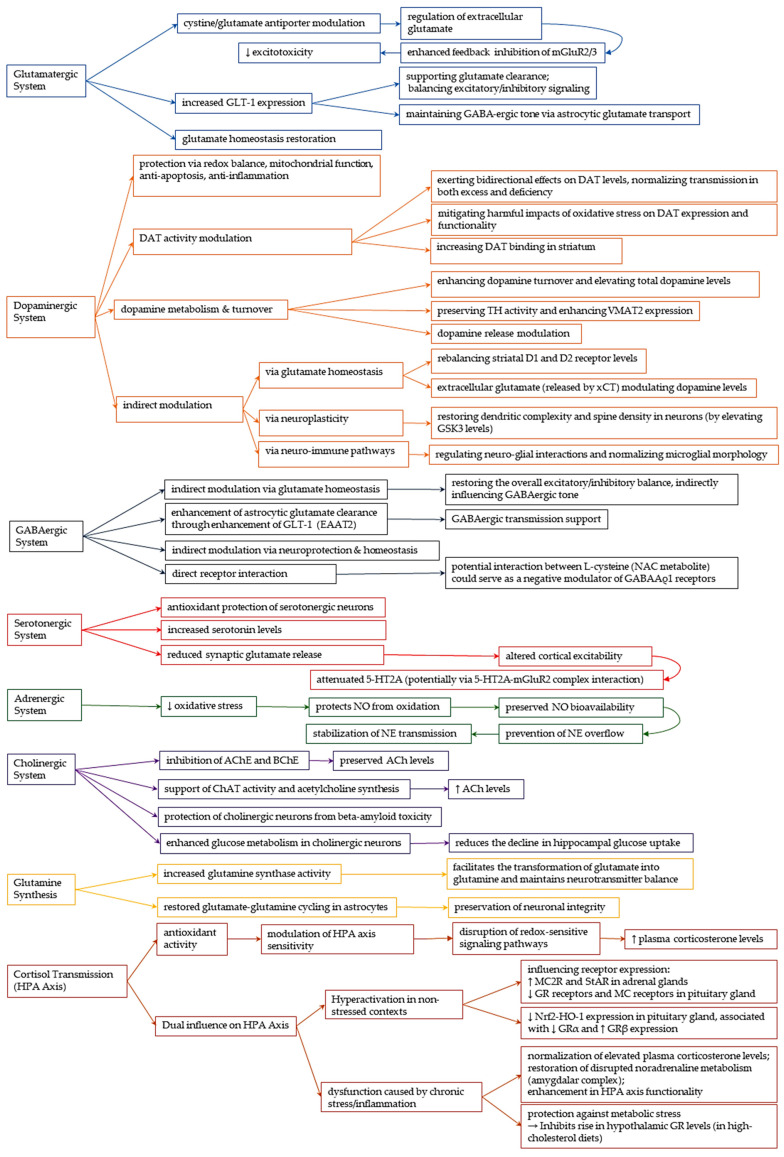
Neuromodulatory properties of NAC.

**Table 1 cimb-47-00710-t001:** Antioxidant Mechanisms of NAC.

	Mechanism	Key Effects/Outcomes	References
GSH-Dependent	Enhances intracellular GSH synthesis via cysteine precursor	Restores redox balance; protects neurons in Parkinson’s disease	[4,18]
	De novo GSH synthesis via GCL and GS	Maintains GSH levels	[30]
	Redox recycling via GPx	Detoxifying reactive peroxides	[4,21]
	Xenobiotic conjugation via GSTs	Detoxifies electrophilic compounds	[4,21]
	Modulates α-synuclein and NFκB signaling	Reduces neuroinflammation and protein aggregation	[25,45]
	Increases brain GSH (animal and some human studies)	Confers neuroprotection	[35]
	Protects against age-related and chemotherapy-induced oxidative damage	Preserves neuronal integrity	[32,33,34,45]
GSH-Independent	Direct antioxidant effects via –SH group	Scavenges NO_2_, hypohalous acids; limited in vivo efficacy	[4,38]
	Neutralization of lipid peroxides (via ALOX5 inhibition)	Prevents ferroptosis, reduces neuronal loss in ICH models	[21]
	Mitochondrial sulfane sulfur/H_2_S production via MST and SQR	Generates hydropersulfides; enhances mitochondrial redox balance	[40]
	Activation of Nrf2-ARE pathway	Upregulates antioxidant and detox genes; improves GSH levels and reduces inflammation	[19,41,42,43,44]
	DNA repair and anti-apoptotic gene modulation	Decreases 8OH2dG; regulates Bax, p53, and Bcl-2 expression	[10]
	Suppression of heme-oxygenase-1 (HO-1)	Limits dopamine-triggered oxidative stress in oligodendrocytes	[28]
	SOD and CAT activity up-regulation	Restores enzymatic antioxidant defenses after chronic stress or cisplatin	[26,33]
	Inhibition of oxidative stress–induced MAPK activation	Prevents tau hyperphosphorylation and aggregation Preserves neuronal structure and function	[29,46,47]
	Modulating the Cysteine Redox Proteome	Addresses dysregulation of cysteine residues, prevents protein aggregation	[37]

**Table 2 cimb-47-00710-t002:** Anti-Neuroinflammatory Mechanisms of NAC.

Mechanism	Pathway/Target	Effect of NAC	Key Findings/Studies
Suppression of Cytokine Production and NF-κB/iNOS Inhibition	TNF-α, IL-1β, IL-6, NF-κB, iNOS	↓ Cytokines ↓ iNOS expression ↓ NF-κB activation ↑ GSH levels	- ↓ cerebral infarct size post-stroke [51] - ↓ iNOS in hippocampus [53] - ↓ NF-κB and cytokines in stress model [33]
PPAR-α Activation	PPAR-α-Peroxisomal proteins	↑ PPAR-α activity ↑ OL development ↑ Peroxisomal function	- NAC restores OL development via PPAR-α [55] - GSH replenishment and NF-κB inactivation help restore PPAR-α
JAK/STAT Pathway Modulation	JAK1/2/3, Tyk2, STAT3, SOCS3	↓ STAT3 mRNA/protein ↑ SOCS3 expression ↓ Phosphorylation of STAT3	- NAC inhibits JAK/STAT in depression model [56] - NAC may glutathionylate STAT3, blocking activation
Microglial Modulation	Connexins and Pannexins MMP-3/MMP-9 ROS, TNF-α	↓ Pro-inflammatory cytokines ↓ ROS ↓ MMPs Modulates microglial activity	- Connexin/pannexin modulation affects neuroinflammation [11] - Dose-dependent microglial effects: low dose = protective, high dose = toxic [57] - ↓ MMPs and cytokines in LPS models [59]
Restoration of PVI Functionality	MMP9/RAGE-Perineuronal nets GABA signaling	↑ PVI network integrity ↓ RAGE ↑ Prefrontal GABA Improved cognition	- Sequential NAC + Environmental Enrichment (EE) in schizophrenia model [58] - Clinical trial in early psychosis shows cognitive improvement

**Table 3 cimb-47-00710-t003:** Mechanisms of Enhanced Neuroprotection and Cellular Viability by NAC.

Submechanism	Pathway/Target	Effect of NAC	Key Findings/Studies
Initiation of Pro-survival Signaling			
ERK Pathway Activation	Cdk5-p35 ERK Bcl-2	↑ Cdk5 and ERK ↑ Bcl-2 ↓ Aβ toxicity	- Prevents Aβ-induced apoptosis in neurons [61] - Also involved in ammonia toxicity mitigation [9]
Inhibition of MAPK and Tau Phosphorylation	JNK, p38 Tau (Ser199, Ser202, Thr205, Ser396)	↓ Tau phosphorylation ↓ JNK/p38 activity	- Reduced H_2_O_2_-induced tau aggregation [29] - Inhibits MAPK in hippocampal neurons [47]
Mitochondrial Protection and Redox Balance			
Mitochondrial Dynamics Regulation	CDK5-DRP1 GPx1-NF-κB Parvalbumin	↓ DRP1 fission ↑ GPx1 ↑ ATP and mitochondrial integrity	- Protects inhibitory interneurons in epilepsy model [62]
Blood–Brain Barrier Modulation	Occludin, Claudin-5 nNOS Connexins 37/43	Time-dependent BBB effects: 24 h ↑ Permeability 48 h ↑ Integrity	- Cell-specific, redox- and NOS-dependent effects [63]
Redox Balance and Energy Metabolism	NADPH cycling Complex I	↑ Mitochondrial respiration ↓ ROS (astrocytes)	- Coordinated control of junctions and mitochondria in BBB [63]
Modulation of Neurotrophic Factors			
	BDNF TrkB PI3K/Akt, ERK, PLC-γ	↑ BDNF expression ↑ ERK signaling ↓ Apoptosis	- Supports neurogenesis and plasticity [18] - Restores ERK1/2 and mitochondrial function in cadmium toxicity [64]

## Data Availability

Not applicable.

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
