# Peer review of "The Central Nervous System Modulatory Activities of N-Acetylcysteine: A Synthesis of Two Decades of Evidence"

_cimb, 2025, doi:10.3390/cimb47090710_

Round 1

Reviewer 1 Report

Comments and Suggestions for Authors

This article provides a review of the literature to evaluate the neuromodulating mechanisms of NAC and its therapeutic potential in various diseases of the central nervous system. This extensive review article summarizes the results of preclinical and clinical trials and highlights promising areas of research for its use in medicine.  The following comments arose during the review.

  1. The authors' desire to provide a systematic review of this issue over a fairly long period of time is understandable, but it must be clearly understood that the literature must not only be analyzed, but also carefully selected so as not to overload the article with a mass of minor details that interfere with the perception of the main idea of the review. Perhaps the authors should shorten the review, mainly by eliminating historical essays in the study of the problem.
  2. The Introduction section, although it contains generalizing information, needs to be more clearly emphasized regarding the priorities that the authors have identified as the main ones for further detailed consideration in the review article.
  3. Section 3.1 includes well-known data on oxidative metabolism in the nervous system. This section of the review is poorly organized and overloaded with information. To emphasize the involvement of NAC, the authors may well provide a scheme for its inclusion in the metabolism of glutathione. This can clearly and concretely demonstrate the involvement of NAC in oxidative metabolism and will contribute to the reduction and concretization of this section.
  4. Section 3.1.2. is much better structured than section 3.1. It is also necessary to highlight the subsections in this section in Section 3.1.
  5. The Nrf2-ARE Pathway Activation section also contains a lot of descriptive, unstructured information. It would be highly advisable to present this biological pathway in the diagram, indicating the involvement of NAC in the activation process.
  6. Section 3.2. Anti-inflammatory effects also needs to be improved. Currently, there are many reviews on this topic, for example, "Shabab, T.; Khanabdali, R.; Moghadamtousi, S.Z.; Kadir, H.A.; Mohan, G. Neuroinflammation Pathways: A General Review. Int. J. Neurosci. 2017, 127, 624-633., in which the information is presented much more clearly and in an orderly manner than in section 3.2. of this review.
  7. Due to the poor study of the material, the sections are very different in volume, for example, section 3.2.2 is described very briefly, while sections 3.1 and 3.2.3, on the contrary, are unnecessarily overblown. Sections 3.2.3 and 3.3 also need to improve the style of writing and systematization of the presented material.
  8. In section 3.3.1, perhaps the authors should provide a diagram that clearly illustrates the involvement of NAC in the Ras extracellular signal-regulated kinase pathway. This section is in particular need of editing and stylistic correction of the English language, as well as systematization and generalization of the presented data.
  9. Sections 3.3.2 and 3.3.3 need stylistic correction.
  10. Section 3.4 Modulation of neurotransmission is unnecessarily overblown, it needs to be shortened and more detailed study of the material. Sections 3.4.8 and 3.5 look unnecessarily schematic compared to Section 3.4. The authors need to systematize the information presented, rather than leaving it in the form of a vast stream of undifferentiated data that is poorly developed by the authors and unnecessarily bloated.
Comments on the Quality of English Language

English needs correction

Author Response

Answers to Reviewer 1

This article provides a review of the literature to evaluate the neuromodulating mechanisms of NAC and its therapeutic potential in various diseases of the central nervous system. This extensive review article summarizes the results of preclinical and clinical trials and highlights promising areas of research for its use in medicine.  The following comments arose during the review.

  1. The authors' desire to provide a systematic review of this issue over a fairly long period of time is understandable, but it must be clearly understood that the literature must not only be analyzed, but also carefully selected so as not to overload the article with a mass of minor details that interfere with the perception of the main idea of the review. Perhaps the authors should shorten the review, mainly by eliminating historical essays in the study of the problem.

We express our gratitude to the reviewer for this insightful and constructive feedback. We completely concur that the meticulous selection of literature sources is crucial for enhancing the clarity and overall significance of the review. In this context, we have made revisions to the manuscript, specifically in the Introduction and Section 3.1, to eliminate historical or peripheral information that does not directly pertain to the core theme—namely, the neuroprotective mechanisms of NAC. This revision has facilitated a more streamlined structure and has better aligned the manuscript with the reviewer’s suggestions.

Additionally, we wish to clarify that the current manuscript is not designed as a systematic review; instead, it serves as a narrative synthesis of the accumulated data regarding NAC-related neuromodulatory mechanisms over the last 25 years. Our primary aim is to encapsulate the specific pathways through which NAC affects the central nervous system, particularly emphasizing its potential implications in neurological and psychiatric disorders.

  1. The Introduction section, although it contains generalizing information, needs to be more clearly emphasized regarding the priorities that the authors have identified as the main ones for further detailed consideration in the review article.

We express our gratitude to the reviewer for this insightful suggestion. We completely concur that the Introduction ought to more distinctly delineate the priorities and scope of the review. In response to this, we have incorporated a succinct paragraph at the conclusion of the Introduction that explicitly articulates the emphasis on the neuromodulatory mechanisms of NAC:

Over the past two decades, extensive research has examined the actions of N-acetylcysteine in the CNS. In this review, we prioritize synthesizing mechanistic evidence from both preclinical and clinical studies conducted during the last 25 years. Rather than covering general pharmacological effects, we examine how NAC affects neuromodulatory pathways that contribute to its therapeutic effects. These include its antioxidant and anti-inflammatory actions, effects on neuroprotection and cell viability, modulation of neurotransmitter systems, and its influence on gene expression. The review is structured thematically, aiming to organize complex findings in a way that highlights specific molecular and cellular pathways relevant to neurological and psychiatric disorders.”

Furthermore, we have condensed the Introduction by eliminating redundant statements and minimizing background details regarding the extensive medical uses of NAC. Sections of the initial text were substituted with more succinct expressions, facilitating a more straightforward progression towards the primary aim of the review—examining the mechanisms by which NAC influences the central nervous system.

  1. Section 3.1 includes well-known data on oxidative metabolism in the nervous system. This section of the review is poorly organized and overloaded with information. To emphasize the involvement of NAC, the authors may well provide a scheme for its inclusion in the metabolism of glutathione. This can clearly and concretely demonstrate the involvement of NAC in oxidative metabolism and will contribute to the reduction and concretization of this section.

We are grateful for the constructive feedback provided by the reviewer. We recognize the concern regarding the excessive general information present in Section 3.1 and concur that this section needed to be streamlined. Consequently, we have minimized the content pertaining to general oxidative metabolism that does not directly relate to NAC, thereby enhancing the focus and conciseness of the section.

In accordance with the reviewer’s valuable suggestion, we have also included a schematic illustration that depicts the metabolic integration of NAC within the glutathione synthesis pathway. This figure is supplemented by a brief explanatory text that emphasizes the direct role of NAC in essential antioxidant mechanisms. We believe that this visual representation greatly enhances the clarity and specificity of the section, while also highlighting the mechanistic significance of NAC in oxidative metabolism.

  1. Section 3.1.2. is much better structured than section 3.1. It is also necessary to highlight the subsections in this section in Section 3.1.

We express our gratitude to the reviewer for the constructive suggestion concerning the structural arrangement of Section 3.1. We acknowledge that the recommendation pertains mainly to the introductory portion of this section—specifically, subsection 3.1.1.

Section 3.1 is organized around two primary categories of antioxidant mechanisms: glutathione-dependent (3.1.1) and glutathione-independent (3.1.2) pathways. In response to the reviewer’s feedback and in line with the clearer structure already established in subsection 3.1.2, we have restructured subsection 3.1.1 by incorporating clearly defined subheadings. These subheadings categorize the specific processes in which NAC is engaged through its influence on the glutathione system.

Considering the volume and nature of the pertinent content, the most effective approach to enhance the structure in subsection 3.1.1 was to divide the discussion into two principal sections: preclinical and clinical evidence. We have also improved the logical progression of the text to enhance readability. Furthermore, a schematic figure has been included to depict the role of NAC within glutathione metabolism. This visual component aims to clarify and highlight NAC’s contribution to these antioxidant pathways.

We are confident that these revisions have greatly enhanced the clarity and focus of Section 3.1 and have effectively addressed the reviewer’s concerns regarding information overload and structural deficiencies.

  1. The Nrf2-ARE Pathway Activation section also contains a lot of descriptive, unstructured information. It would be highly advisable to present this biological pathway in the diagram, indicating the involvement of NAC in the activation process.

We are grateful for the reviewer’s valuable recommendation. Following this suggestion, we have incorporated a schematic diagram that encapsulates the essential steps involved in the activation of the Nrf2-ARE pathway and demonstrates NAC’s role in this process—specifically, through the activation of Nrf2, its translocation to the nucleus, and the subsequent transcription of ARE-dependent antioxidant genes. To enhance the figure, we have also revised the related text in Section 3.1.2 to improve its organization and focus. The updated section now clearly delineates the sequential progression of the pathway and underscores the mechanisms through which NAC may promote its activation. Additionally, we have enriched the conclusion of this section with an extra paragraph that emphasizes a GSH-independent pathway of Nrf2 modulation by NAC: “Beyond serving as a GSH precursor, NAC may activate Nrf2 indirectly via hydrogen sulfide (H₂S). By acting as a substrate for 3-mercaptopyruvate, NAC increases mitochondrial H₂S levels, which can persulfonate cysteine residues on Keap1, impairing its ability to sequester Nrf2 in the cytosol. This post-translational modification of Keap1 releases Nrf2, allowing its nuclear translocation and activation of antioxidant gene expression, providing a GSH-independent route for NAC-driven Nrf2 activation. Upon activation, Nrf2 increases the expression of major antioxidant systems, including GSH, GSSG, glutathione peroxidase, glutathione S-transferases, thioredoxin (Trx), and thioredoxin reductase (TrxR). The Trx system further sustains Nrf2 activity through a positive feedback loop. Nrf2 also boosts NADPH production by upregulating metabolic enzymes and rerouting glucose metabolism toward the pentose phosphate pathway (PPP). Additionally, it enhances NAD(P)H:quinone oxidoreductase 1 (NQO1) expression, which supports mitochondrial function and lowers oxidative and nitrosative stress.” (Morris et al.)

We are confident that this revised section is now more coherent and accurately represents the mechanistic significance of NAC in modulating the Nrf2-ARE pathway, through both traditional and alternative molecular pathways.

  1. Section 3.2. Anti-inflammatory effects also needs to be improved. Currently, there are many reviews on this topic, for example, "Shabab, T.; Khanabdali, R.; Moghadamtousi, S.Z.; Kadir, H.A.; Mohan, G. Neuroinflammation Pathways: A General Review. Int. J. Neurosci. 2017, 127, 624-633., in which the information is presented much more clearly and in an orderly manner than in section 3.2. of this review.

We express our gratitude to the reviewer for the insightful comment and for directing us to the thorough review by Shabab et al. (2017), which has been referenced in our manuscript. This article offers a detailed examination of the primary neuroinflammatory mediators and their signaling pathways, which may serve as potential therapeutic targets in neurodegenerative processes.

In light of the reviewer’s suggestion and to enhance the clarity and readability of Section 3.2, we have incorporated Table 2, which outlines the established mechanisms that underlie the anti-neuroinflammatory effects of NAC. This table organizes the information in a more structured and accessible manner, emphasizing key molecular targets and pathways involved, along with pertinent evidence from both preclinical and clinical studies.

We are confident that this addition improves the organization and informative quality of the section, while still keeping the focus of our review on the NAC-specific neuromodulatory effects..

  1. Due to the poor study of the material, the sections are very different in volume, for example, section 3.2.2 is described very briefly, while sections 3.1 and 3.2.3, on the contrary, are unnecessarily overblown. Sections 3.2.3 and 3.3 also need to improve the style of writing and systematization of the presented material.

We express our gratitude to the reviewer for this significant observation. The variation in length among the sections largely reflects the current state of research concerning the respective mechanisms. As mentioned, Sections 3.2.2 and 3.2.3 address mechanisms that are supported by a relatively limited number of studies, which accounts for their more succinct presentation. Conversely, Sections 3.1 and 3.2.3 encompass pathways that have undergone more extensive investigation, leading to a more thorough discussion.

In response to the reviewer’s feedback, we have revised the style and enhanced the organization of Sections 3.2.3 and 3.3 to improve clarity and consistency throughout the manuscript. Additionally, we have included a new table that summarizes the available findings on NAC’s effects on neuronal viability and survival. This addition aims to systematize the data and offers a clearer visual representation of the evidence presented in the text.

We are confident that these modifications enhance the overall balance, readability, and informativeness of the review.

  1. In section 3.3.1, perhaps the authors should provide a diagram that clearly illustrates the involvement of NAC in the Ras extracellular signal-regulated kinase pathway. This section is in particular need of editing and stylistic correction of the English language, as well as systematization and generalization of the presented data.

We are grateful for the reviewer’s valuable feedback. We completely concur that Section 3.3.1 necessitated considerable revision. In response, we have meticulously revised the entire section to enhance the English language, clarity of expression, and overall style. Additionally, we have reorganized the content to ensure a more logical and coherent presentation of the material.

The updated version now commences with a concise overview of the fundamental components and signaling principles of the Ras/ERK (extracellular signal-regulated kinase) pathway, followed by a more concentrated discussion of the specific modulatory effects of NAC, as documented in the existing literature. This approach aids in contextualizing NAC’s role within the pathway and offers a clearer framework for interpreting the data.

We are confident that these revisions have greatly enhanced the section’s structure, readability, and scientific significance.

  1. Sections 3.3.2 and 3.3.3 need stylistic correction.

We thank the reviewer for this comment. In response, we have carefully revised Sections 3.3.2 and 3.3.3 to improve the style, clarity, and consistency of the language. We believe these corrections have enhanced the overall readability and flow of the text in these subsections..

  1. Section 3.4 Modulation of neurotransmission is unnecessarily overblown, it needs to be shortened and more detailed study of the material. Sections 3.4.8 and 3.5 look unnecessarily schematic compared to Section 3.4. The authors need to systematize the information presented, rather than leaving it in the form of a vast stream of undifferentiated data that is poorly developed by the authors and unnecessarily bloated.

We express our gratitude to the reviewer for this significant observation. As noted in earlier sections, the variation in both length and depth across the subsections of Section 3.4 mirrors the current landscape of research—certain neuromodulatory pathways of NAC have been studied in greater detail, while others are still relatively unexplored. This variation accounts for the more extensive detail found in some areas of the section, such as 3.4, in contrast to others like 3.4.8 and 3.5.

In light of the reviewer’s feedback, we have made revisions to Section 3.4 to enhance the text's clarity and eliminate redundancies. Furthermore, we have reorganized the content to facilitate better clarity and ensure a more systematic presentation of the data.

Moreover, to address the concern regarding the schematic representation, we wish to highlight that a comprehensive integrative figure has been added at the conclusion of this section. This diagram consolidates the existing data on NAC’s neuromodulatory effects across various neurotransmitter systems and aids in visually organizing the mechanisms outlined in the text.

We are confident that these modifications significantly enhance the focus, coherence, and informativeness of this section.

Dear Reviewer 1,

We genuinely value the thorough feedback and constructive recommendations offered by the reviewer, which have significantly enhanced our manuscript. We have meticulously evaluated each suggestion and implemented the necessary revisions throughout the document.

We are confident that the modifications made have positively influenced the quality of our review.

Reviewer 2 Report

Comments and Suggestions for Authors The manuscript cimb-3790468, entitled "The Central Nervous System Modulatory Activities of N-acetylcysteine", reviews many matters of the action of NAC on the nervous system. However, despite the extensive description, the manuscript lacks focus and seems not to add but few things that could be new to the current knowledge in the field. It is rather a very long descriptive work on the NAC mechanisms of action. Moreover, very few sentences could be identified as interpretative work of the narrated information, and the manuscript generally lacks the critical point of view of the Authors.

Author Response

The manuscript cimb-3790468, entitled "The Central Nervous System Modulatory Activities of N-acetylcysteine", reviews many matters of the action of NAC on the nervous system. However, despite the extensive description, the manuscript lacks focus and seems not to add but few things that could be new to the current knowledge in the field. It is rather a very long descriptive work on the NAC mechanisms of action. Moreover, very few sentences could be identified as interpretative work of the narrated information, and the manuscript generally lacks the critical point of view of the Authors.

We are grateful to Reviewer 2 for taking the time to review our article, as well as for making recommendations to improve its quality.

In response to his comments, we would like to clarify that the initial goal of our review was not a critical review, but rather a summary of the available information (a narrative review) on the neuroprotective mechanisms of the substance established so far, especially since they appear to be so numerous and diverse. We believe that collecting the data in one place, as well as the summary schemes, would be useful for the readers of this article, because it allows for a quick and easy reference on this topic. We would like to take the liberty of including here a portion of the cover letter justifying the choice of topic and format of the article: "Several reviews are available from recent years that address the biological properties of this medicine. For instance, the recent analysis conducted by Martinez-Banaclocha from 2022 (https://doi.org/10.3390/antiox11020416) offers extensive insights into the molecular mechanisms that contribute to neurodegenerative diseases. It particularly highlights the dysregulation of the cysteine redox proteome and the antioxidant mechanism by which NAC demonstrates its neuroprotective properties. Conversely, the review also explores the potential of N-acetyl-cysteine as an antioxidant and a GSH-boosting agent, highlighting its capacity to cross the blood-brain barrier and influence critical cysteine-containing proteins within the brain. The paper presents both preclinical and clinical evidence supporting its effectiveness across a range of conditions, such as Alzheimer's disease, Parkinson's disease, and amyotrophic lateral sclerosis.

Unlike other materials that emphasize a particular neuroprotective mechanism, our review does not limit itself to the well-known antioxidant mechanism of neuroprotection. Instead, it seeks to include all other recognized mechanisms that contribute positively to neuronal health in the context of neurodegenerative processes as a whole. The objective of our paper is to consolidate the recognized neuroprotective properties of N-acetylcysteine. It appears to modulate various cell-signaling pathways and demonstrate multiple mechanisms of action that may help mitigate neuronal damage and dysfunction, pointing it out as a promising, well-tolerated candidate for neurotherapeutic applications. Its pleiotropic mechanisms—spanning antioxidant, anti-inflammatory, neuromodulatory, and gene-regulatory pathways—support its use in a range of neurodegenerative and psychiatric disorders."

To respond to the comments made by Reviewer 2, we included additional section named Critical Interpretation after the Discussion.

We genuinely hope that the enhancements implemented effectively enhance the quality of this review and address the recommendations made by the Reviewer.

Round 2

Reviewer 1 Report

Comments and Suggestions for Authors

The authors have significantly improved the content of the review article in accordance with the recommendations made. A revised version of the article after English correction may be recommended for publication.

Comments on the Quality of English Language

A revised version of the article after English correction may be recommended for publication.